# THE STATE OF REINFORCEMENT FINETUNING FOR TRANSFORMER-BASED AGENTS

**Shengchao Hu[1,2,†], Peng Wang[3,†], Guozheng Ma[2], Shi Fu[2], Li Shen[3], Ya Zhang[1], Dacheng Tao[2*]**
[1] Shanghai Jiao Tong University, [2] Nanyang Technological University,
[3] Shenzhen Campus of Sun Yat-sen University
{charles-hu,ya_zhang}@sjtu.edu.cn; wangp389@mail.sysu.edu.cn;
{GUOZHENG001,shi011}@e.ntu.edu.sg; {mathshenli,dacheng.tao}@gmail.com

## ABSTRACT

Reinforcement finetuning (RFT) has garnered significant attention in recent years, particularly for enhancing large reasoning models such as OpenAI o1 and Deepseek R1. The appeal of RFT largely stems from its ability to refine model knowledge, better align outputs with user intent, and address challenges associated with limited finetuning data. Despite these advantages, the application of RFT in large Transformer-based agents remains relatively underexplored. Although these agents are designed to address multiple tasks through large-scale autoregressive pretraining and share many properties with large reasoning models, current adaptation strategies predominantly rely on supervised finetuning (SFT). In this work, we conduct a systematic investigation of several RFT techniques across a variety of finetuning parameter configurations and meta-reinforcement learning (meta-RL) environments, employing few-shot offline datasets. We provide a comprehensive analysis of RFT algorithm performance under diverse experimental conditions and, based on our empirical findings, introduce a lightweight enhancement to existing RFT methods. This enhancement consistently improves outcomes by combining the strengths of both SFT and RFT. Our findings provide valuable insights for advancing the effectiveness of RFT approaches and broadening their applicability to meta-RL tasks with large Transformer-based agents, motivating further research in broader domains.

## 1 INTRODUCTION

Large Reasoning Models (LRMs), such as OpenAI o1 (Jaech et al., 2024) and DeepSeek R1 (Guo et al., 2025), represent the leading edge of artificial intelligence, characterized by advanced reasoning and extended deliberation capacities. A key development in these models is the adoption of reinforcement finetuning (RFT)(Lambert et al., 2024; Team et al., 2025), which facilitates efficient adaptation to domain-specific tasks with limited labeled data. Through iterative refinement—particularly in test-time scaling—RFT enhances reasoning ability, factual accuracy, and alignment with user intent and ethical standards(Kumar et al., 2025). Consequently, RFT has become integral to both the advancement and deployment of large-scale AI systems.

The paradigm of leveraging and transferring knowledge from large-scale pretrained models within LRMs has catalyzed substantial progress across a wide range of domains (Dosovitskiy et al., 2020; Raffel et al., 2020). Within reinforcement learning (RL), Transformer-based Agents (TAs) have demonstrated exceptional efficacy and generalization by formulating decision-making problems as sequence modeling tasks (Wen et al., 2022; Reed et al., 2022). These agents employ Transformer-based autoregressive decoders, which facilitate multi-modality, multi-tasking, and scalable general-purpose decision-making (Chen et al., 2021; Janner et al., 2021). Notably, TAs exhibit robust generalization to novel tasks through zero-shot and few-shot trajectory conditioning, and share many properties with LRMs (Hu et al., 2024c; Agarwal et al., 2023). However, despite these similarities, the application of RFT for adapting TAs to new tasks—particularly in the context of meta-RL—remains

---

*Corresponding author: Dacheng Tao
†These authors contributed equally to this work.

underexplored. Current approaches for TA adaptation predominantly rely on supervised finetuning (SFT) (Wang et al., 2024; Hu et al., 2026), which may limit generalization to new tasks. Meanwhile, RL-based finetuning methods have achieved notable success in non-RL domains, suggesting their potential advantages for RL tasks, especially those with dense rewards (Schulman et al., 2017). These observations motivate our investigation into whether RL-based finetuning can outperform SFT for adapting TAs to novel RL tasks, and to systematically evaluate its effectiveness in this context.

In this paper, we present a systematic study of RFT for TAs, with a particular emphasis on two prominent finetuning paradigms: (1) full-model finetuning (FMT), which updates all model parameters, and (2) parameter-efficient finetuning (PET), which optimizes only a small subset of (often newly introduced) parameters, such as prompt tuning (Hu et al., 2026), low-rank adaptation (LoRA)(Hu et al., 2021), and the decorator approach(Yuan et al., 2024). We further investigate a range of finetuning algorithms, including: (1) supervised methods such as SFT (Wang et al., 2024; Hu et al., 2026) and DPO (Rafailov et al., 2023), (2) online RL algorithms such as GRPO (Guo et al., 2025; Shao et al., 2024) and PPO (Schulman et al., 2017), and (3) offline RL methods such as CQL (Kumar et al., 2020a). Building on our experimental findings, we also propose a lightweight Q-guided Policy Optimization (QP) that integrates the advantages of SFT with RFT approaches. To this end, we conduct extensive experiments to systematically evaluate the interplay between finetuning algorithms and finetuning parameter configurations, providing a comprehensive analysis of their effectiveness for adapting TAs to new RL tasks.

Additionally, we provide in-depth analyses of several key factors that may influence finetuning performance. Specifically, we examine the impact of (1) the quality of the finetuning dataset, (2) the quantity of available finetuning trajectories, (3) the prevalence of sparse versus dense reward signals, and (4) the scale of the pretrained model. By systematically evaluating these variables, we aim to elucidate how different finetuning algorithms and parameter adaptation strategies perform under varying conditions, thereby offering comprehensive insights into their robustness and generalization capabilities across diverse RL scenarios. Our main contributions are summarized below:

- We introduce and rigorously analyze the application of RFT to TAs in meta-RL settings, evaluating performance across diverse finetuning parameter configurations and environments.

- We propose a lightweight enhancement to existing RFT methods by integrating SFT with RFT-based policy updates, resulting in robust and consistent performance improvements.

- We conduct extensive empirical studies across various tasks, systematically investigating factors affecting finetuning outcomes, including finetuning trajectory quality, finetuning dataset size, reward sparsity, and pretrained model scale. Our proposed method demonstrates significant improvements over strong SFT and RFT baselines in all settings.

## 2 BACKGROUND

### 2.1 TRANSFORMER-BASED AGENT

The adoption of Transformer architectures (Vaswani et al., 2017) in offline RL for sequential modeling (SM) has garnered significant attention in recent years (Hu et al., 2024c). Insights from NLP demonstrate that Transformers, when pre-trained on large-scale datasets, exhibit remarkable few-shot and zero-shot learning capabilities within prompt-based frameworks (Liu et al., 2023b; Brown et al., 2020). Inspired by these findings, recent studies have sought to develop sufficiently large agents capable of tackling diverse tasks (Reed et al., 2022; Wen et al., 2022). Many of these works leverage the architectural principles of the prompt-based Decision Transformer (DT) (Xu et al., 2022; Hu et al., 2026), which adapts prompt-based methodologies from NLP to the offline RL setting, thereby enabling few-shot generalization to novel tasks.

Unlike NLP, where prompts are typically textual and often formatted for blank-filling to adapt to various tasks, Prompt-DT introduces trajectory prompts. These prompts comprise tuples of state, action, and return-to-go $(\mathbf{s}^*, \mathbf{a}^*, \hat{r}^*)$, offering explicit guidance to RL agents through a small number of demonstration steps. Each element with a superscript $\cdot^*$ denotes its association with the trajectory prompt. Notably, the length of a trajectory prompt is typically much shorter than the full task horizon, encapsulating only the essential information required for effective task identification, yet insufficient for complete imitation of the task.

During pretraining with offline collected data $\mathcal{D}_i$, Prompt-DT utilizes $\tau_{i,t}^{input} = (\tau_i^*, \tau_{i,t})$ as input for each task $\mathcal{T}_i$. Here, $\tau_{i,t}^{input}$ consists of the $K^*$-step trajectory prompt $\tau_i^*$ and the most recent $K$-step history $\tau_{i,t}$, and is formulated as:

$$\tau_{i,t}^{input} = (\hat{r}_{i,1}^*, \mathbf{s}_{i,1}^*, \mathbf{a}_{i,1}^*, \ldots, \hat{r}_{i,K^*}^*, \mathbf{s}_{i,K^*}^*, \mathbf{a}_{i,K^*}^*,$$
$$\hat{r}_{i,t-K+1}, \mathbf{s}_{i,t-K+1}, \mathbf{a}_{i,t-K+1}, \ldots, \hat{r}_{i,t}, \mathbf{s}_{i,t}, \mathbf{a}_{i,t}). \tag{1}$$

The prediction head linked to a state token $\mathbf{s}$ is designed to predict the corresponding action $\mathbf{a}$. For continuous action spaces, the training objective aims to minimize the mean-squared loss:

$$\mathcal{L}_{DT} = \mathbb{E}_{\tau_{i,t}^{input} \sim \mathcal{D}_i} \left[ \frac{1}{K} \sum_{m=t-K+1}^{t} (\mathbf{a}_{i,m} - \pi(\tau_i^*, \tau_{i,m}))^2 \right]. \tag{2}$$

In contrast to LLM-based agent frameworks (Park et al., 2023), where the model interacts with the environment through natural-language actions, the TAs considered in our work operate in continuous-control domains: they consume structured state representations and generate continuous action sequences, and their performance is evaluated by task-specific reward signals.

## 2.2 REINFORCEMENT FINETUNING

With the advent of advanced reasoning models such as OpenAI's o1 (Jaech et al., 2024), research on large reasoning models has increasingly focused on enhancing reasoning capabilities through RL techniques. Recent studies have demonstrated improved performance in reasoning-intensive tasks, including mathematical problem-solving (Cai et al., 2024; Trung et al., 2024; Shao et al., 2024; Yang et al., 2024) and code generation (Hui et al., 2024; Jiao et al., 2024; Zhang et al., 2024a;b). A notable milestone is Deepseek-R1-Zero (Guo et al., 2025), which achieved robust reasoning abilities using RL alone, entirely omitting the SFT stage. However, most RL-based reasoning research has been restricted to the language domain, with limited investigation into Transformer-based agents in meta-RL settings, where traditional SFT remains the standard for finetuning (Zhao et al., 2022). To address this gap, our work systematically analyzes the efficacy of RFT for such agents.

## 3 MODEL FINETUNING OVERVIEW

### 3.1 FINETUNING ALGORITHMS

Given a pretrained agent $\pi_\theta$ and corresponding few-shot finetuning datasets $\mathcal{P}_i$ for each task $\mathcal{T}_i$—where $|\mathcal{P}_i| \ll |\mathcal{D}_i|$ (the dataset used for pretraining)—various strategies can be employed to further adapt the policy via reinforcement learning. For simplicity, although the transformer-based policy $\pi$ typically conditions on full history, i.e., $\pi_\theta(\mathbf{a}_t|\hat{r}_{:t}, \mathbf{s}_{:t}, \mathbf{a}_{:t-1})$, here we denote it as $\pi_\theta(\mathbf{a}_t|\mathbf{s}_t)$. Note that finetuning need not update all parameters; further details are discussed in Section 3.2. For brevity, we use $\theta$ to denote the set of parameters subject to tuning in this context. Due to practical constraints, we could not include all prominent RFT algorithms in our experiments and therefore select a representative subset of widely used methods.

**Supervised Fine-Tuning (SFT)** is the most straightforward way to adapt a pretrained model. It minimizes the mean squared error (MSE) between the predicted and ground-truth actions on the small finetuning dataset $\mathcal{P}$:

$$\mathcal{L}_{\text{SFT}}(\theta) = \mathbb{E}_{(\mathbf{s},\mathbf{a})\sim\mathcal{P}} \left[ |\mathbf{a} - \pi_\theta(\mathbf{a}|\mathbf{s})|^2 \right]. \tag{3}$$

**Direct Preference Optimization (DPO)** (Rafailov et al., 2023) reformulates reinforcement learning from human feedback (RLHF) as a direct optimization problem. For each state $\mathbf{s}$, preference pairs $(\mathbf{a}, \bar{\mathbf{a}})$ are constructed, with $\bar{\mathbf{a}}$ representing a perturbed version of $\mathbf{a}$. The policy is optimized to favor preferred actions while maintaining proximity to a reference model $\pi_{\text{ref}}$, with $\beta$ controlling the trade-off:

$$\mathcal{L}_{\text{DPO}}(\theta) = -\mathbb{E}_{(\mathbf{s},\mathbf{a},\bar{\mathbf{a}})\sim\mathcal{P}} \left[ \log \sigma \left( \beta \log \frac{\pi_\theta(\mathbf{a}|\mathbf{s})}{\pi_{\text{ref}}(\mathbf{a}|\mathbf{s})} - \beta \log \frac{\pi_\theta(\bar{\mathbf{a}}|\mathbf{s})}{\pi_{\text{ref}}(\bar{\mathbf{a}}|\mathbf{s})} \right) \right], \tag{4}$$

where $\sigma$ is the logistic function.

**Group Relative Policy Optimization (GRPO)** (Shao et al., 2024; Guo et al., 2025) compares groups of candidate responses directly, eliminating the need for an additional critic model. Here we consider the trajectories with the same initial state as a group of candidate responses. The objective encourages higher probabilities for preferred actions, regularized with a KL divergence to the reference policy:

$$\mathcal{L}_{\text{GRPO}}(\theta) = -\mathbb{E}_{(\mathbf{s},\{\mathbf{a}^k\}_{k=1}^K)\sim\mathcal{P}} \tag{5}$$

$$\left[\sum_{i=1}^K (\min(\frac{\pi_\theta(\mathbf{a}^k|\mathbf{s})}{\pi_{\theta_{\text{old}}}(\mathbf{a}^k|\mathbf{s})})A_k, \text{clip}(\frac{\pi_\theta(\mathbf{a}^k|\mathbf{s})}{\pi_{\theta_{\text{old}}}((\mathbf{a}^k|\mathbf{s}))}, 1-\epsilon, 1+\epsilon)A_k) - \beta D_{\text{KL}}(\pi_\theta||\pi_{ref}))\right], \tag{6}$$

$$D_{\text{KL}}(\pi_\theta||\pi_{ref}) = \frac{\pi_{ref}(\mathbf{a}^k|\mathbf{s})}{\pi_\theta(\mathbf{a}^k|\mathbf{s})} - \log\frac{\pi_{ref}(\mathbf{a}^k|\mathbf{s}_i)}{\pi_\theta(\mathbf{a}^k|\mathbf{s}_i)} - 1, \tag{7}$$

where $\epsilon$ and $\beta$ are hyper-parameters, and $A_k$ is the advantage, computed using a group of rewards $\{r_1, r_2, \ldots, r_K\}$ corresponding to the outputs within each group:

$$A_k = \frac{r_k - \text{mean}(\{r_1, r_2, \ldots, r_K\})}{\text{std}(\{r_1, r_2, \ldots, r_K\})}. \tag{8}$$

**Proximal Policy Optimization (PPO)** (Schulman et al., 2017) is a widely used policy gradient algorithm. It maximizes a clipped surrogate objective to ensure stable updates by constraining the policy within a trust region around the previous policy $\pi_{\text{old}}$:

$$\mathcal{L}_{\text{PPO}}(\theta) = -\mathbb{E}_{(\mathbf{s},\mathbf{a})\sim\mathcal{P}}\left[\min\left(r_\theta(\mathbf{a}|\mathbf{s})\hat{A}, \text{clip}\left(r_\theta(\mathbf{a}|\mathbf{s}), 1-\epsilon, 1+\epsilon\right)\hat{A}\right)\right], \tag{9}$$

where $r_\theta(\mathbf{a}|\mathbf{s}) = \frac{\pi_\theta(\mathbf{a}|\mathbf{s})}{\pi_{\text{old}}(\mathbf{a}|\mathbf{s})}$ is the probability ratio between the current and old policies, and $\hat{A}$ denotes the advantage estimate, which is calculated by the learned critic network with generalized advantage estimation (Schulman et al., 2015). Note that, as PPO is an on-policy method, its effectiveness is limited in strictly offline settings, but the PPO loss can still be used for gradient computation.

**Conservative Q-Learning (CQL)** (Kumar et al., 2020b) is designed for offline RL, mitigating overestimation by penalizing Q-values for out-of-distribution actions. The loss function combines a Bellman loss with a conservative regularization term:

$$\mathcal{L}_{\text{CQL}}(\phi) = \mathbb{E}_{(\mathbf{s},\mathbf{a},r,\mathbf{s}')\sim\mathcal{P}}\left[\frac{1}{2}\left(Q_\phi(\mathbf{s},\mathbf{a}) - (r + \gamma\mathbb{E}_{\mathbf{a}'\sim\pi_\theta}Q_\phi(\mathbf{s}',\mathbf{a}'))\right)^2\right] \tag{10}$$

$$+ \alpha\left(\mathbb{E}_{\mathbf{a}\sim\pi_\theta}[Q_\phi(\mathbf{s},\mathbf{a})] - \mathbb{E}_{\mathbf{a}\sim\mu}[Q_\phi(\mathbf{s},\mathbf{a})]\right), \tag{11}$$

where $\mu$ denotes the random policy and $\alpha$ is the regularization strength. After learning the Q-function, policy improvement is performed by updating the policy $\pi_\theta$ to maximize the expected Q-value, which can be formulated as:

$$\mathcal{L}_{\text{CQL}}(\theta) = -\mathbb{E}_{\mathbf{s}\sim\mathcal{P}}\left[Q_\phi(\mathbf{s}, \pi_\theta(\mathbf{s}))\right]. \tag{12}$$

Here, the policy parameters $\theta$ are optimized to select actions that maximize the value predicted by the learned Q-network, thereby iteratively improving the policy while remaining robust to the limitations of the offline dataset.

**Q-guided Policy Optimization (QP, ours).** In RFT methods such as PPO and CQL, a Q-network is often used to estimate the advantage function. However, in offline RL settings, Q-network estimates are often unreliable due to distributional shift, necessitating additional constraints or regularization to prevent divergence from the behavior policy. Conversely, supervised methods can maintain policy stability but are generally limited to in-distribution actions, which may lead to sub-optimal performance. Recent studies have demonstrated the potential of extending Q-learning to Transformer-based agents for offline RL, achieving promising results (Hu et al., 2024a). Building on this line of research and the detailed results in Sections 5 and 6, we propose Q-guided policy optimization (QP), which augments the objective of supervised methods by incorporating a learned Q-network. Specifically, we augment standard supervised training (e.g., SFT or DPO) with a policy improvement term guided by the Q-network. This approach, denoted as QP-SFT or QP-DPO, aims to combine

the distributional robustness of supervised methods with the policy improvement capabilities of RL, thereby enhancing both stability and performance in offline settings:

$$\mathcal{L}_{\text{QP-SFT}}(\theta) = \mathbb{E}_{(\mathbf{s},\mathbf{a})\sim\mathcal{P}}\left[|\mathbf{a} - \pi_\theta(\mathbf{a}|\mathbf{s})|^2 - \alpha \cdot Q_\phi(\mathbf{s}, \pi_\theta(\mathbf{s}))\right], \tag{13}$$

$$\mathcal{L}_{\text{QP-DPO}}(\theta) = \mathbb{E}_{(\mathbf{s},\mathbf{a},\bar{\mathbf{a}})\sim\mathcal{P}}\left[\log\sigma\left(\beta\log\frac{\pi_\theta(\mathbf{a}|\mathbf{s})}{\pi_{\text{ref}}(\mathbf{a}|\mathbf{s})} - \beta\log\frac{\pi_\theta(\bar{\mathbf{a}}|\mathbf{s})}{\pi_{\text{ref}}(\bar{\mathbf{a}}|\mathbf{s})}\right) - \alpha \cdot Q_\phi(\mathbf{s}, \pi_\theta(\mathbf{s}))\right], \tag{14}$$

where $\alpha$ controls the regularization strength and $Q_\phi$ is trained via a TD-loss with double Q-learning (Hasselt, 2010):

$$\mathbb{E}_{(\mathbf{s},\mathbf{a},\mathbf{s}')\sim\mathcal{P}}\left\lVert\hat{Q}_m - Q_{\phi_i}(\mathbf{s},\mathbf{a})\right\rVert^2, \tag{15}$$

$$\text{where} \quad \hat{Q}_m = r + \gamma\min_{i=1,2}Q_{\phi_i'}(\mathbf{s}',\hat{\mathbf{a}}), \tag{16}$$

where $\gamma$ is the discount factor and $\hat{a}$ denotes the predicted action output by the target model $\pi_{\theta'}$. The first term in Equations 13–14 corresponds to the standard imitation (SFT or DPO) objective, anchoring the updated policy to the behavior distribution. The second term provides a value-based correction that encourages the policy to select actions with higher predicted return under $Q_\phi$. In this way, QP augments the stability of supervised learning with conservative value-guided policy improvement, enabling the policy to extrapolate beyond the behavior policy while mitigating the risk of distributional drift in offline settings.

## 3.2 FINETUNING PARAMETER CONFIGURATIONS

In the context of finetuning large Transformer-based agents for RL, various parameter-efficient and full-model adaptation strategies can be considered. Below, we detail several approaches for finetuning different components of the agent, as illustrated in Figure 6 and further described in Section D.

**Prompt Tuning.** This approach updates only the prompt parameters—typically initialized from sampled task trajectories $\mathcal{P}$—while keeping the backbone model fixed (Hu et al., 2026; 2025b). Prompt tuning enables efficient few-shot adaptation with minimal risk of overfitting.

**Adaptor Tuning.** Following Huang et al. (2024), this method inserts lightweight LoRA modules, typically into the MLP layers of the Transformer (Lawson & Qureshi, 2024; Hu et al., 2024e). Only the adaptor parameters are updated for each new task, allowing efficient and isolated adaptation without affecting the shared backbone.

**Decorator Tuning.** Inspired by residual policy learning (Yuan et al., 2024), this strategy introduces a residual policy $\pi_{\text{res}}$ trained on top of a frozen base policy $\pi_{\text{base}}$. The action taken is the sum of both policies: $\pi_{\text{base}}(s) + \pi_{\text{res}}(s)$, enabling targeted adaptation while retaining prior knowledge.

**Fullmodel Tuning.** All parameters of the agent are finetuned jointly. While this maximizes adaptation capacity, it increases computational cost and overfitting risk in low-data regimes.

## 4 EXPERIMENTAL SETUP

**Environments.** We evaluate all proposed methods on two standard meta-RL benchmarks: (i) the multi-task MuJoCo locomotion suite (Ni et al., 2023; Todorov et al., 2012), which includes the Cheetah-Dir, Cheetah-Vel, and Ant-Dir; and (ii) the MetaWorld robotic manipulation platform (Yu et al., 2020), comprising 50 distinct tasks. For the MuJoCo locomotion suite, we follow the protocol of Wang et al. (2024) by randomly sampling tasks from the overall distribution and partitioning them into a training set ($\mathcal{T}^{\text{train}}$) and a test set ($\mathcal{T}^{\text{test}}$). In MetaWorld, 45 tasks are designated for pretraining, while the remaining 5 tasks are reserved for meta-testing and adaptation. To construct the datasets, we employ Soft Actor-Critic (SAC)(Haarnoja et al., 2018) to independently train single-task policies for each training task. We then generate two types of offline datasets—Medium and Expert—corresponding to different levels of policy proficiency. Further details regarding the environments and dataset construction are provided in Appendix B.

**Training.** We instantiate Prompt-DT as our TA backbone, implemented on top of the open-source `minGPT` codebase. For each environment, we first pretrain a policy on the offline dataset $\mathcal{D}$,

Table 1: Performance of different finetuning algorithms on meta-RL environments using expert-level offline datasets. Each method is evaluated with 50 finetuning trajectories, and results are averaged over three independent runs with different random seeds.

| | Zero-shot | SFT | DPO | GRPO | PPO | CQL | QP-DPO | QP-SFT |
|---|---|---|---|---|---|---|---|---|
| **Prompt** | | | | | | | | |
| AntDir | 259.61 | **326.52**$_{\pm3.62}$ | 319.45$_{\pm3.03}$ | 323.07$_{\pm3.11}$ | 316.53$_{\pm2.72}$ | 325.93$_{\pm3.50}$ | 325.14$_{\pm12.77}$ | 325.35$_{\pm3.78}$ |
| HalfCheetahDir | 600.49 | 631.01$_{\pm5.46}$ | 634.60$_{\pm1.73}$ | 638.38$_{\pm4.87}$ | 629.12$_{\pm1.12}$ | 628.52$_{\pm1.43}$ | **645.24**$_{\pm3.89}$ | 639.36$_{\pm2.04}$ |
| HalfCheetahVel | −138.01 | −111.57$_{\pm2.40}$ | −113.37$_{\pm0.46}$ | −113.79$_{\pm3.79}$ | **−109.66**$_{\pm2.90}$ | −114.72$_{\pm0.66}$ | −111.06$_{\pm1.20}$ | −110.64$_{\pm2.23}$ |
| MetaWorld | 414.22 | 414.22$_{\pm0.00}$ | 414.22$_{\pm0.00}$ | 414.22$_{\pm0.00}$ | 414.22$_{\pm0.00}$ | 414.22$_{\pm0.00}$ | 414.22$_{\pm0.00}$ | 414.22$_{\pm0.00}$ |
| Average | 284.08 | 315.05 | 313.73 | 315.47 | 312.55 | 313.49 | 318.39 | 317.07 |
| **Adaptor** | | | | | | | | |
| AntDir | 259.61 | 315.06$_{\pm3.62}$ | 325.81$_{\pm5.62}$ | 319.66$_{\pm3.60}$ | 335.57$_{\pm9.71}$ | 332.05$_{\pm6.45}$ | **336.62**$_{\pm2.87}$ | 329.36$_{\pm6.43}$ |
| HalfCheetahDir | 600.49 | 631.06$_{\pm1.97}$ | 637.39$_{\pm4.12}$ | 632.83$_{\pm3.37}$ | 633.84$_{\pm0.83}$ | 636.30$_{\pm2.01}$ | 631.91$_{\pm2.47}$ | **640.60**$_{\pm2.73}$ |
| HalfCheetahVel | −138.01 | −118.98$_{\pm2.52}$ | −115.32$_{\pm3.93}$ | **−109.74**$_{\pm3.98}$ | −113.51$_{\pm0.93}$ | −119.11$_{\pm1.54}$ | −113.22$_{\pm3.47}$ | −112.19$_{\pm1.19}$ |
| MetaWorld | 414.21 | 499.60$_{\pm24.56}$ | 485.63$_{\pm15.21}$ | 490.21$_{\pm38.21}$ | 454.91$_{\pm22.21}$ | 437.28$_{\pm30.10}$ | **502.52**$_{\pm4.76}$ | 485.71$_{\pm5.74}$ |
| Average | 284.08 | 331.69 | 333.38 | 333.24 | 327.70 | 321.63 | 339.46 | 335.87 |
| **Decorator** | | | | | | | | |
| AntDir | 259.61 | 320.63$_{\pm2.83}$ | 332.66$_{\pm9.81}$ | 317.25$_{\pm6.65}$ | 304.08$_{\pm17.93}$ | **354.51**$_{\pm6.94}$ | 323.07$_{\pm5.36}$ | 330.87$_{\pm3.75}$ |
| HalfCheetahDir | 600.49 | 632.33$_{\pm2.80}$ | 639.85$_{\pm4.19}$ | 629.89$_{\pm4.43}$ | 636.70$_{\pm2.47}$ | 637.53$_{\pm4.09}$ | 637.10$_{\pm1.76}$ | **640.64**$_{\pm2.33}$ |
| HalfCheetahVel | −138.01 | −118.46$_{\pm3.42}$ | −121.67$_{\pm2.92}$ | **−108.52**$_{\pm3.76}$ | −117.13$_{\pm5.30}$ | −109.65$_{\pm2.10}$ | −113.20$_{\pm2.54}$ | −112.90$_{\pm3.46}$ |
| MetaWorld | 414.21 | 452.42$_{\pm27.06}$ | 407.78$_{\pm5.42}$ | 413.52$_{\pm8.36}$ | 468.47$_{\pm25.46}$ | 408.91$_{\pm2.71}$ | 473.35$_{\pm24.34}$ | **478.03**$_{\pm14.92}$ |
| Average | 284.08 | 321.73 | 314.66 | 313.04 | 323.03 | 322.83 | 330.08 | 334.16 |
| **Fullmodel** | | | | | | | | |
| AntDir | 259.61 | 401.63$_{\pm12.31}$ | 360.96$_{\pm2.33}$ | 319.69$_{\pm3.22}$ | 335.06$_{\pm10.50}$ | 304.95$_{\pm10.96}$ | 370.08$_{\pm2.45}$ | **412.18**$_{\pm16.48}$ |
| HalfCheetahDir | 600.49 | 634.61$_{\pm1.28}$ | 643.79$_{\pm3.67}$ | 630.08$_{\pm0.69}$ | 634.82$_{\pm4.20}$ | 627.78$_{\pm3.14}$ | **650.84**$_{\pm4.01}$ | 642.62$_{\pm3.96}$ |
| HalfCheetahVel | −138.01 | −133.02$_{\pm4.15}$ | −122.31$_{\pm4.63}$ | −119.87$_{\pm4.03}$ | −120.37$_{\pm4.45}$ | −117.22$_{\pm3.44}$ | −121.39$_{\pm1.60}$ | **−119.79**$_{\pm0.26}$ |
| MetaWorld | 414.21 | 441.05$_{\pm14.92}$ | 424.88$_{\pm27.26}$ | 471.02$_{\pm23.65}$ | 468.97$_{\pm24.53}$ | 434.59$_{\pm13.35}$ | 548.98$_{\pm30.80}$ | **553.36**$_{\pm35.35}$ |
| Average | 284.08 | 336.07 | 326.83 | 325.23 | 329.62 | 312.53 | 362.13 | 372.09 |

which aggregates trajectories from multiple distinct tasks. The detailed model architecture and hyperparameters are provided in Appendix C. Subsequently, for each finetuning algorithm, we initialize from the same pretrained policy and apply the four finetuning parameter configurations described in Section 3.2, utilizing the same finetuning trajectory set $\mathcal{P}$ ($|\mathcal{P}| = 50$) and maintaining identical finetuning iterations to ensure a fair comparison. Notably, the number of trajectories available for finetuning is substantially smaller than that used during pretraining ($|\mathcal{P}| \ll |\mathcal{D}|$, corresponding to only 0.1%–1.1% of the pretraining data), thereby establishing a few-shot adaptation setting that more closely reflects practical, real-world scenarios. For each loss function, all methods are tuned via grid search over the corresponding hyperparameters to ensure that the reported results represent their best achievable performance. For each method, we select and report the best-performing checkpoint over training, which accounts for potentially different convergence dynamics while preserving fairness in the comparison.

## 5 THE STATE OF REINFORCEMENT FINETUNING

Here, we show the empirical results for reinforcement finetuning in the pretrained transformer-based agents in Table 1 and Figure 8. Additional experimental results, theoretical support, and extended discussions are provided in Appendix E and Appendix F.

**The effect of finetuning parameters.** We compare four finetuning parameter configurations—Prompt (0.76KB), Adaptor (0.19MB), Decorator (0.54MB), and Fullmodel (2.52MB)—to investigate the impact of parameter count on finetuning efficacy. Experimental results across meta-RL environments indicate that finetuning performance is highly sensitive to the choice of parameter configurations. While Fullmodel finetuning—updating the largest parameter set—achieves the highest average scores for both QP-DPO and QP-SFT, it is not uniformly optimal. Performance exhibits a clear method-dependent preference: supervised approaches (SFT, DPO) generally benefit from full-model updates, whereas RL-based methods (e.g., GRPO, CQL) often perform best under parameter-efficient schemes (Adapters, Decorators). This dichotomy reflects differences in gradient supervision and signal propagation across learning paradigms. In supervised settings, direct imitation of provided trajectories or preferences is most effectively achieved through full-model finetuning, which facilitates comprehensive signal propagation across all network layers and typically yields stronger convergence and generalization (Mandlekar et al., 2021). Conversely, RL finetuning operates on inherently noisy, high-variance reward signals, where update errors compound over extended horizons, causing full-model parameter updates to exhibit unstable oscillatory behavior (Kumar et al., 2022). In

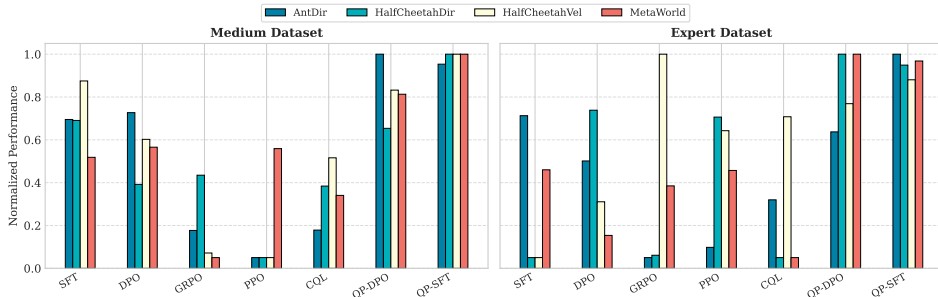

Figure 1: Comparison of finetuning dataset quality across different finetuning algorithms in meta-RL settings, under a fixed finetuning budget of 50 trajectories. Performance is normalized such that the highest score for each setting is scaled to 1.

such regimes, parameter-efficient finetuning restricts the update subspace, mitigating overfitting to stochastic rewards and improving stability and sample efficiency. Moreover, the results suggest that Prompt-based finetuning, while parameter-efficient, tends to underperform in high-variance or complex environments (e.g., MetaWorld), likely due to its limited expressive flexibility. In the MetaWorld setting, the pretrained agent possesses only limited prior knowledge about the unseen downstream tasks, and updating solely the prompt parameters does not provide sufficient capacity to substantially reshape the policy. Consequently, all finetuning objectives are constrained to optimize the same narrowly scoped parameter subset, and thus tend to converge to similar performance levels.

**The effect of finetuning algorithms.** Empirical results across diverse meta-RL environments and parameter configurations consistently demonstrate the superiority of QP-based finetuning algorithms (QP-DPO, QP-SFT) over both supervised methods (SFT, DPO) and RL-based approaches. QP variants frequently achieve the highest or near-highest performance across various tasks and finetuning parameters, highlighting their robustness and broad applicability. By optimizing policies with respect to soft value estimates or preference-guided advantages, QP methods combine the stability and sample efficiency of supervised learning with the adaptability and asymptotic performance of RL finetuning. In contrast, pure RL methods such as PPO and CQL, while occasionally competitive, often exhibit instability and lower average performance. This is likely attributable to the lack of strong inductive priors from imitation-based objectives and the high variance in gradient estimates, particularly in few-shot demonstration regimes. Notably, in certain tasks—such as MetaWorld with Adaptor finetuning parameter—the choice of finetuning algorithm can yield greater performance improvements than merely increasing finetuning parameter size. For example, with PPO, switching the finetuning parameter from Adaptor to Decorator results in a modest 3% performance gain, whereas changing the algorithm to QP-DPO yields a 10% improvement. These findings underscore the critical importance of algorithm selection in meta-RL settings.

> **Takeaway**: No single algorithm consistently achieves optimal performance across all finetuning parameter configurations and environments, underscoring the need for adaptive finetuning strategies. Moreover, the choice of finetuning algorithm has a substantial impact on overall performance, often comparable to or greater than the effect of finetuning parameter scale.

## 6 DISCUSSION

**Ablation on finetuning data quality and quantity.** We jointly analyze the effects of dataset quality (Medium vs. Expert) and the number of finetuning trajectories on seven algorithms (as shown in Figure 1 and Figure 2). Two consistent patterns emerge. First, *quality*: QP-DPO and QP-SFT dominate across both medium- and expert-quality regimes. RL-based methods (e.g., GRPO, PPO) exhibit pronounced gains under expert data, suggesting stronger sensitivity to high-quality demonstrations and effective credit assignment when optimal trajectories are present. In contrast, supervised methods benefit relatively more from medium-quality datasets that contain a larger proportion of sub-optimal trajectories, providing a stable imitation-based lower bound on policy performance. Second, *quantity*: increasing the number of finetuning trajectories typically improves performance for all methods, underscoring the centrality of data availability for policy adaptation. Notably, QP-based methods

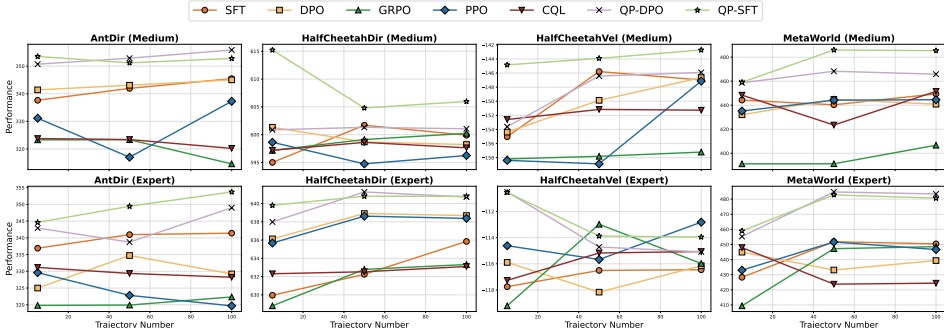

Figure 2: Impact of the number of finetuning trajectories on different algorithms across various environments, evaluated with both Medium and Expert datasets. Scores are averaged over multiple finetuning parameter configurations for each algorithm within each environment.

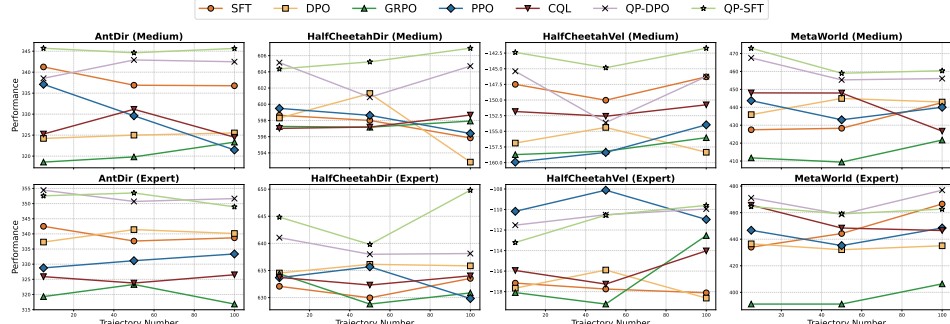

Figure 3: Impact of the number of finetuning trajectories on the performance of various algorithms across multiple environments, assessed under both Medium and Expert dataset conditions in the sparse reward setting. Scores are computed as the average across multiple finetuning parameter configurations for each respective finetuning algorithm within the environment.

sustain superior results even in low-data settings, while purely supervised approaches degrade more sharply when trajectories are scarce, indicating higher sample sensitivity.

> **Takeaway**: (i) Supervised approaches yield stable, imitation-driven performance—particularly when data quality is mixed and the number of trajectories exceeds a moderate scale—whereas RL-based methods can surpass them given expert demonstrations and are comparatively advantageous in low-data regimes; (ii) Increasing the number of finetuning trajectories does not universally lead to performance improvements, the effect depends on the choice of finetuning algorithm and environment; and (iii) Across regimes of both quality and quantity, QP-based methods (QP-DPO, QP-SFT) exhibit mostly improvement trends, showing the robustness.

**Ablation study of the sparse reward setting.** To assess the robustness of learning algorithms under challenging feedback conditions, we evaluate model performance in sparse reward environments, where agents receive only a cumulative reward at the final timestep—mirroring recent RL finetuning practices in LLM reasoning (Guo et al., 2025; Swamy et al., 2025). As shown in Figure 3, sparse rewards significantly widen performance disparities both across algorithms and data regimes. QP-DPO and QP-SFT exhibit greater robustness, maintaining stable performance with minimal degradation, which we attribute to their policy optimization mechanisms that effectively combine the strengths of SFT and RFT. In contrast, GRPO suffers from persistent underperformance, likely due to optimization challenges and insufficient sample efficiency in offline settings. On the other hand, we occasionally observe that RFT methods achieve higher performance under sparse rewards than under dense rewards. This does not contradict the usual benefits of dense rewards in interactive RL, but reflects a specific failure mode in our low-data offline regime. In the dense case, TD-style value and advantage estimates aggregate bootstrapped targets at every timestep; with very few trajectories and poor state–action coverage, approximation errors of the learned value function at intermediate states are repeatedly propagated backward, leading to severely biased advantages. In the sparse construction, we keep the same total return but concentrate it at the terminal step, so the value target is dominated by a single, well-defined terminal return. Under data scarcity, this makes the update

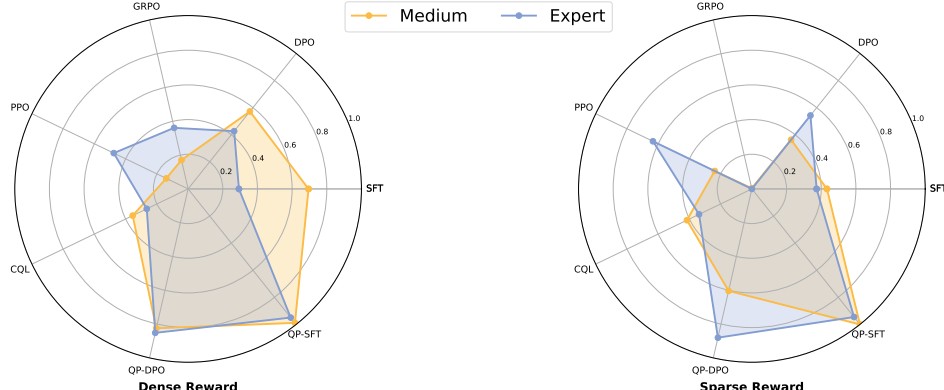

Figure 4: Performance comparison of various methods under both sparse and dense reward conditions. Each method is evaluated with 50 finetuning trajectories, and results are averaged across multiple environments and a range of finetuning parameter configurations.

more Monte Carlo–like, reduces sensitivity to value-function approximation error, and can yield relatively less noisy and more stable advantage estimates.

Data quality proves particularly critical in the sparse regime: models trained on expert data consistently incur less performance loss than those trained on medium data, highlighting the value of optimal demonstrations when reward signals are limited. Moreover, while the relative trends between supervised and RL-based approaches are similar to those observed in dense reward settings, their magnitudes diverge. In low-data sparse settings, RL-based methods (e.g., PPO) demonstrate smaller improvements as the number of trajectories increases, indicating that the absence of dense feedback limits the benefits of additional data and constrains policy improvement. Conversely, supervised approaches show more stable and monotonic gains, as their objective is insensitive to reward sparsity. In high-quality settings, however, RFT methods continue to achieve comparable asymptotic performance, reflecting their capacity to exploit optimal trajectories more effectively.

A comparative summary across dense and sparse reward conditions (Figure 4) reinforces these observations. QP-based methods remain consistently superior and robust across feedback structures, with their performance margin over alternative approaches more pronounced in the sparse than in the dense setting. By contrast, GRPO attains competitive results primarily under dense rewards but degrades markedly under sparse supervision, a decline plausibly attributable to inaccurate group-advantage estimation in the absence of frequent intermediate feedback.

> **Takeaway**: In sparse reward environments, (i) data quality plays a pivotal role—expert trajectories substantially mitigate performance degradation; (ii) RFT methods achieve higher asymptotic returns given high-quality data, reflecting their ability to exploit optimal demonstrations; (iii) supervised approaches provide more stable gains across data regimes, being less sensitive to reward sparsity; and (iv) QP-based methods demonstrate superior robustness across both dense and sparse settings, confirming their adaptability to different feedback structures.

**Ablation study of scaling pretrained agents.** We examine the effect of model scaling on downstream finetuning performance in the MetaWorld benchmark by systematically comparing four finetuning parameter configurations under both Medium and Expert data regimes. In certain configurations, such as Adaptor and Fullmodel, the number of trainable parameters increases proportionally with the size of the pretrained agent, whereas in others, such as Prompt and Decorator, the parameter count remains constant regardless of model size. Additional details are provided in Appendix F.

Empirical results in Figure 5 indicate that most finetuning strategies benefit from larger pretrained agent sizes, exhibiting consistent performance improvements as model capacity increases. However, in the case of Fullmodel finetuning, the substantial rise in trainable parameters associated with larger models—combined with limited finetuning data—often results in performance plateaus or even degradation. These findings underscore the necessity of balancing model capacity with the availability of high-quality supervision when developing scalable finetuning strategies.

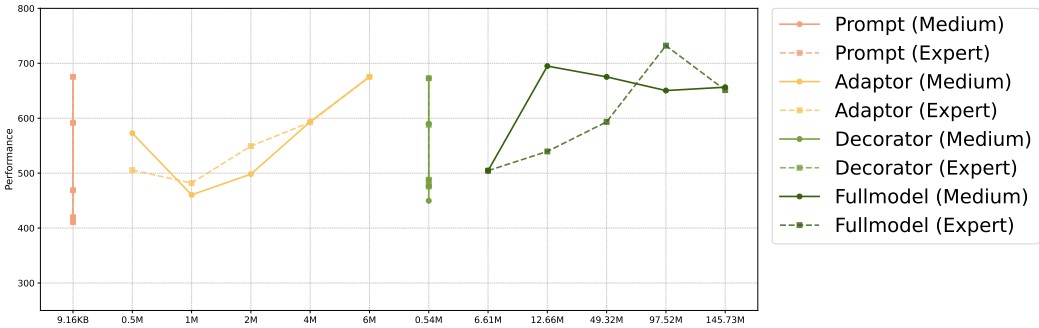

Figure 5: Performance of QP-DPO under various finetuning parameter configurations across different pretrained agent sizes in the MetaWorld benchmark. For certain configurations (e.g., Adaptor, Fullmodel), the number of trainable parameters increases with the size of the pretrained agent, while for others, the parameter count remains constant.

> **Takeaway**: Training instability and overfitting can arise when the parameter space is expanded without sufficient data, as the resulting optimization problem becomes increasingly challenging to solve effectively.

## 7 CONCLUSION

In this work, we investigate the state of RFT for TAs in meta-RL settings. From an empirical perspective, we evaluated the performance of various RFT methods across a range of finetuning parameter configurations within meta-RL environments. Furthermore, we systematically examined the impact of key factors—including the quality and quantity of finetuning data, reward sparsity, and pretrained model scale—on downstream performance. Additionally, we introduce a lightweight extension that integrates SFT with RFT-based policy improvement. This approach mostly demonstrated superior results across all evaluated scenarios.

We envision this study as a foundation for future research on RFT algorithms for TAs, particularly in real-world applications (e.g., robotics, autonomous driving, and safety-critical domains) where online exploration is expensive or risky. In such settings, collecting a small set of high-quality expert trajectories is often more practical than running large-scale online RL. By combining supervised imitation with value-guided policy improvement, QP-based methods help bridge offline RL, imitation learning, and LLM-style post-training for embodied and multi-modal agents, thereby providing a principled and practical pathway toward robust TA adaptation in complex environments.

**Limitation.** Although this work investigates RFT for pretrained TAs, the largest models evaluated have up to 40 million parameters—considerably smaller than contemporary large language models with hundreds of millions or billions of parameters. While sizable relative to traditional RL agents, this scale may not fully capture the behaviors of larger models. Future work incorporating larger agents may yield different insights into the effectiveness of RFT.

ACKNOWLEDGMENTS

Dr Tao's research is partially supported by NTU RSR and Start Up Grants. Li Shen is supported by National Key R&D Projects (NO. 2024YFC3307100), NSFC Grant (No. 62576364), Shenzhen Basic Research Project (Natural Science Foundation) Basic Research Key Project (NO. JCYJ20241202124430041), CCF-DiDi GAIA Collaborative Research Funds (NO. CCF-DiDi GAIA 202508).

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

**The Use of Large Language Models.** In this work, we exclusively employ large language models (LLMs) to refine the writing and presentation of our manuscript.

## A  RELATED WORK

**Offline Reinforcement Learning.** Offline RL trains policies solely from static datasets, without further environment interaction (Levine et al., 2020). This is especially valuable when interactions are costly or risky. However, offline RL faces challenges from distribution shift between the behavior and learned policies, often leading to performance drops (Fujimoto et al., 2019). Approaches such as constrained or regularized dynamic programming help mitigate this issue (Fujimoto & Gu, 2021; Kumar et al., 2020a; Kostrikov et al., 2021). Conditional sequence modeling offers a supervised alternative, predicting actions from past state-action-reward sequences and keeping the learned policy close to the data distribution (Chen et al., 2021; Hu et al., 2025a; 2024a; Yamagata et al., 2023; Hu et al., 2026; 2024d; Meng et al., 2023). This LLM-inspired paradigm enables scalable RL with large data and compute. Diffusion models have also been adopted for offline RL, using generative modeling techniques to represent policies or dynamics and achieving strong empirical results (Janner et al., 2022; Ajay et al., 2022a; Chen et al., 2022; Wang et al., 2022). Related work by Kumar et al. (2022) examines when offline RL outperforms behavioral cloning (supervised fine-tuning), addressing a question closely aligned in our motivation. Their focus is standard offline RL benchmarks and conditions under which policy improvement surpasses imitation, whereas our focus is *meta-RL*: a pretrained, Transformer-based agent must adapt to *unseen* tasks using only few-shot, offline data. The two perspectives are therefore complementary—ours emphasizes rapid cross-task adaptation under strict data constraints, while theirs characterizes the policy-improvement vs. imitation trade-off within a single-task offline regime.

**Offline Meta-Reinforcement Learning.** Offline meta-RL seeks to generalize to new tasks by training on a distribution of offline tasks (Gao et al., 2023; Ni et al., 2023). Optimization-based methods (Finn et al., 2017; Xu et al., 2018; Mitchell et al., 2021) and context-based approaches (Rakelly et al., 2019; Zintgraf et al., 2021; Li et al., 2020; Yuan & Lu, 2022; Gao et al., 2023; Li et al., 2024) have both been explored, typically relying on temporal difference learning. However, these methods often face instability due to the "deadly triad" and depend on hand-crafted heuristics to remain within the offline dataset distribution (Brandfonbrener et al., 2022; Wang et al., 2021; Ajay et al., 2022b). From the perspective of finetuning, Schweighofer et al. (2022) study dataset quality metrics in offline RL, quantifying trajectory returns and state–action coverage of behavior policies and relating these statistics to the performance of classical offline RL algorithms trained from scratch. Schmied et al. (2023) propose the Learning-to-Modulate (L2M) framework, which explores parameter-efficient finetuning and prompt-based tuning for continual-task RL in benchmarks such as MetaWorld, DMControl, and Continual-World, with the goal of mitigating catastrophic forgetting while adapting pretrained models to new tasks. In this work, we focus on RFT for pretrained transformer-based agents in meta-reinforcement learning tasks. This approach contrasts with previous studies that primarily seek to modify model architectures or introduce novel optimization methods.

## B  ENVIRONMENTS AND DATASETS

In this section, we show details of evaluation environments over a variety of testbeds, as well as the offline dataset collection process conducted on these environments.

Following established practice in offline meta-RL (Yuan & Lu, 2022; Mitchell et al., 2021; Wang et al., 2024), we adopt two classical benchmarks: multi-task MuJoCo control (Ni et al., 2023; Todorov et al., 2012) and MetaWorld (Yu et al., 2020). All methods are evaluated on the following environments:

- **MuJoCo Multi-Task Control:** We use Cheetah-Vel, Cheetah-Dir, and Ant-Dir, where task variations arise from differing reward functions.

  - Cheetah-Vel requires a planar cheetah to achieve target velocities sampled uniformly from $U[0.075, 3.0]$, with rewards based on proximity to the goal velocity.
  - Cheetah-Dir and Ant-Dir involve controlling a cheetah or quadruped ant to move in specific directions, with rewards proportional to the cosine similarity between the velocity and goal

Table 2: TQ and SACo of finetuning datasets with 50 trajectories per task.

| Environments | Quality | TQ | SACo |
|---|---|---|---|
| AntDir | Expert | 1.00 | 0.24 |
| AntDir | Medium | 0.41 | 0.27 |
| HalfCheetahDir | Expert | 0.99 | 0.27 |
| HalfCheetahDir | Medium | 0.53 | 0.13 |
| HalfCheetahVel | Expert | 0.99 | 0.50 |
| HalfCheetahVel | Medium | 0.21 | 0.09 |
| MetaWorld | Expert | 1.00 | 0.04 |
| MetaWorld | Medium | 0.34 | 0.06 |

direction (sampled from $U[0, 2\pi]$ for Ant-Dir and limited to forward/backward for Cheetah-Dir). Each episode is limited to 200 steps.

- **MetaWorld:** MetaWorld comprises 50 diverse robotic manipulation tasks with shared dynamics, where a Sawyer robot interacts with objects of varying shapes and mechanisms. Tasks differ in state and reward structures, requiring the robot to achieve different goals using a 4-dimensional action space (3D end-effector position and gripper control). Performance is measured by the average success rate across all tasks.

For both MuJoCo and MetaWorld, we use 45 tasks for training and 5 held-out tasks for evaluation.

For the MuJoCo benchmark, we generate offline datasets by training a separate policy for each task using the Soft Actor-Critic (SAC) algorithm (Haarnoja et al., 2018). Policy checkpoints are periodically saved to produce different dataset types:

- **Medium:** Trajectories generated by a policy achieving between one-third and one-half of expert performance.

- **Expert:** Trajectories generated by the final, converged expert policy.

We collect 100 trajectories per task for each dataset type. For MuJoCo, this yields a pretraining corpus of 4,500 trajectories, whereas finetuning is performed with only 50 trajectories—approximately $1.1\%$ of the pretraining data—placing the evaluation firmly in a few-shot adaptation regime.

For MetaWorld, we follow the procedure of Hu et al. (2024b), training task-specific SAC policies to convergence. Offline datasets are constructed by sampling 1 million transitions per task from the SAC replay buffer, capturing the training process up to convergence. We consider two dataset compositions:

- **Medium:** Consists of the first 50% of trajectories (50 million transitions), representing early-stage learning with reduced expert-level behavior.

- **Expert:** Consists of 100 million transitions per task, spanning from random initialization to expert-level performance.

For a fair comparison between data qualities, we subsample 1,000 expert trajectories to match the medium dataset size. The resulting pretraining corpus contains 45,000 trajectories, while finetuning again uses only 50 trajectories—approximately $0.1\%$ of the pretraining data—thus corresponding to an even more stringent few-shot adaptation regime.

We report the TQ and SACo metrics proposed by Schweighofer et al. (2022) to quantitatively characterize the offline finetuning datasets used in our experiments. For each benchmark environment, we compute and average TQ and SACo over the 5 meta-test tasks, using 50 trajectories per task under both *Expert* and *Medium* demonstration regimes. For metric normalization, we use the full collected dataset (Expert + Medium) as the reference buffer (e.g., 200 trajectories for HalfCheetahDir and 2000 trajectories for MetaWorld). In contrast to Schweighofer et al. (2022), we do not sweep random seeds when constructing the datasets; instead, we report single-seed metric values. As summarized in Table 2, Expert datasets consistently lie in a high-TQ regime (TQ $\approx 1.0$) across all environments, whereas Medium datasets exhibit substantially lower TQ (TQ $\approx 0.21$–$0.53$). SACo values are generally low due to the limited number of finetuning trajectories, with particularly low coverage observed in the MetaWorld benchmarks.

Figure 6: Finetuning parameter configurations. The left panel depicts the Transformer-based Agents, while the four panels on the right illustrate the configurations for Prompt Tuning, Adapter Tuning, Decorator Tuning, and Fullmodel Tuning, respectively.

## C   IMPLEMENTATION DETAILS

**Pretrained Agents.** We build our policy as a Transformer-based model, which is based on minGPT open-source code [1]. We vary the number of Transformer layers, hidden dimensions, and attention heads depending on the scale of the pretrained (from (3, 128, 1) to (48, 256, 16)) model.

**Q-networks.** For finetuning algorithms that require state–action value estimation, we instantiate the Q-function as a three-layer multilayer perceptron with Mish activations and a hidden width of 256 units per hidden layer.

**Compute setup.** All finetuning experiments are conducted on a single NVIDIA RTX 4090 GPU. For pretraining, we use four RTX 4090 GPUs in parallel to accelerate training as the scale of the pre-trained agents increases.

## D   FINETUNING PARAMETER CONFIGURATIONS

In Section 3.2, we introduce four distinct finetuning parameter configurations. Here, we elaborate on these approaches in greater detail, as illustrated in Figure 6.

**Prompt Tuning.** Prompt tuning involves updating only a small set of prompt parameters, which are typically initialized from sampled task trajectories $\mathcal{P}$, while keeping the underlying model backbone fixed (Hu et al., 2026; 2025b). This approach enables highly efficient adaptation to new tasks, particularly in few-shot learning scenarios, with minimal risk of overfitting due to the limited number of trainable parameters. Prompt tuning is especially advantageous when computational resources or labeled data are scarce.

**Adapter Tuning.** Following the methodology of Huang et al. (2024), adapter tuning inserts lightweight parameter-efficient modules—such as LoRA modules—primarily into the MLP layers of the Transformer architecture (Lawson & Qureshi, 2024; Hu et al., 2024e). The number and placement of these adapters are typically determined by the structure of the backbone model, e.g., one adapter per MLP layer. During finetuning, only the adapter parameters are updated for each new task, leaving the shared backbone parameters unchanged. This configuration allows for efficient, isolated task adaptation and facilitates continual learning without catastrophic forgetting.

**Decorator Tuning.** Decorator tuning draws inspiration from residual policy learning (Yuan et al., 2024). In this approach, a residual policy $\pi_{\text{res}}$ is trained on top of a frozen base policy $\pi_{\text{base}}$. The agent's action at each state $s$ is computed as the sum of the outputs from both policies: $\pi_{\text{base}}(s) + \pi_{\text{res}}(s)$. This setup enables targeted adaptation to new tasks while preserving the knowledge encoded in the base policy, thereby supporting both stability and flexibility in policy improvement.

**Fullmodel Tuning.** Fullmodel tuning entails updating all parameters of the agent jointly during finetuning. Although this approach offers the greatest capacity for adaptation and optimization, it also carries a significantly higher risk of overfitting, particularly in low-data regimes. Additionally, full-model tuning substantially increases computational requirements, as all model weights are subject to optimization.

---

[1] `https://github.com/karpathy/minGPT`

# E THEORETICAL SUPPORT

We provide theoretical insights to support the observed performance gains of QP-SFT over standard SFT based on Hu et al. (2024a). Specifically, we derive a performance guarantee under a well-defined set of assumptions in the sequential decision-making setting.

Consider a Markov Decision Process (MDP) $\mathcal{M} = (\mathcal{S}, \mathcal{A}, \mathcal{T}, \mathcal{R}, \mu_0)$, where $\mathcal{S}$ is the state space, $\mathcal{A}$ is the action space, $\mathcal{T} : \mathcal{S} \times \mathcal{A} \times \mathcal{S} \to [0, 1]$ is the state transition probability function, $\mathcal{R} : \mathcal{S} \times \mathcal{A} \to \mathbb{R}$ defines the reward function, and $\mu_0 : \mathcal{S} \to [0, 1]$ is the initial state distribution. We now state the following lemma.

**Lemma E.1** (Alignment with respect to the conditioning function (Brandfonbrener et al., 2022)). *Consider an MDP, behavior $\beta$ and conditioning function $f^r$. Let $J^r(\pi) = \mathbb{E}_{\tau \sim \pi}[g^r(\tau)]$, where $g^r(\tau) = \sum_{t=1}^{\mathcal{H}} r_t$. Assume the following:*

1. *Return coverage: $P_\beta(g^r(\tau) = f^r(\mathbf{s}_1)|\mathbf{s}_1) \geq \alpha_{f^r}$ for all initial states $\mathbf{s}_1$.*

2. *Near determinism: $P(r \neq \mathcal{R}(\mathbf{s}, \mathbf{a})$ or $s' \neq \mathcal{T}(\mathbf{s}, \mathbf{a})|\mathbf{s}, \mathbf{a}) \leq \epsilon$ at all $s, a$ for some functions $\mathcal{T}$ and $\mathcal{R}$. Note that this does not constrain the stochasticity of the initial state.*

3. *Consistency of $f^r$: $f^r(\mathbf{s}) = f^r(\mathbf{s}') + r$ for all $\mathbf{s}$.* [2]

*Then*

$$J^r(\pi^*) - J^r(\pi) \leq \epsilon \left( \frac{1}{\alpha_{f^r}} + 3 \right) \mathcal{H}^2, \tag{17}$$

*where $\pi$ is derived from Equation 3, $\pi^*$ is the optimal policy, and $\mathcal{H}$ is the horizon length of episode. Moreover, there exist problems where the bound is tight up to constant factors.*

**Corollary E.2.** *If $\alpha_{f^r} > 0, \epsilon = 0$, and $f^r(s_1) = V^{r*}(s_1)$ for all initial states $s_1$, then $J^r(\pi^*) = J^r(\pi)$.*

*Remark* E.3. The behavior policy $\beta$ specifies the data-generating distribution of the collected dataset, while the conditioning function $f^r$ corresponds to the return-to-go token $\hat{r}$ used in Transformer-based agents. The lemma and corollary imply that, in nearly deterministic environments, and given appropriate conditioning and sufficient data coverage (i.e., the behavior distribution places support on near-optimal trajectories), SFT can recover policies that are near-optimal.

Based on Lemma E.1, we now give the following Theorem:

**Theorem E.4.** *Consider an MDP with binary rewards and costs, behavior policy $\beta$, and conditioning function $f^r$. Let $g^r(\tau) = \sum_{t=1}^{\mathcal{H}} r_t$, $\mathcal{H}$ is the horizon length of episode. Assume the following:*

1. *Return coverage: $P_\beta(g^r(\tau) = f^r(\mathbf{s}_1)|\mathbf{s}_1) \geq \alpha_{f^r}$ for all initial states $\mathbf{s}_1$.*

2. *Near determinism: $P(r \neq \mathcal{R}(\mathbf{s}, \mathbf{a})$ or $\mathbf{s}' \neq \mathcal{T}(\mathbf{s}, \mathbf{a})|\mathbf{s}, \mathbf{a}) \leq \epsilon$ at all $\mathbf{s}, \mathbf{a}$ for some functions $\mathcal{T}$ and $\mathcal{R}$.*

3. *Consistency of $f^r$: $f^r(\mathbf{s}) = f^r(\mathbf{s}') + r$ for all $\mathbf{s}$.*

*For timestep $i$, the probabilities of selecting actions with maximum reward satisfy:*

**Reward Selection**: *$P\{\hat{P}_i^r - P_i^r \geq \sigma_r, \forall i\} \geq 1 - \delta_r$, where $P_i^r$ and $\hat{P}_i^r$ are probabilities under the policies updated by Equation 3 and Equation 13, respectively. With probability at least $(1 - \delta_r)$:*

$$\mathbb{E}_{\tau \sim \pi^*}[g^r(\tau)] - \mathbb{E}_{\tau \sim \hat{\pi}}[g^r(\tau)] \leq \epsilon(\frac{1}{\alpha_{f^r}} + 3)\mathcal{H}^2 - \mathcal{H}\sigma_r.$$

*where $\hat{\pi}$ is derived from Equation 13.*

---

[2]Note this can be exactly enforced (as in prior work) by augmenting the state space to include the cumulative reward observed so far.

*Proof.* We begin with:

$$\mathbb{E}_{\tau\sim\pi^*}[g^r(\tau)] - \mathbb{E}_{\tau\sim\hat{\pi}}[g^r(\tau)] \tag{18}$$

$$= \mathbb{E}_{\tau\sim\pi^*}[g^r(\tau)] - \mathbb{E}_{\tau\sim\pi}[g^r(\tau)] + \mathbb{E}_{\tau\sim\pi}[g^r(\tau)] - \mathbb{E}_{\tau\sim\hat{\pi}}[g^r(\tau)] \tag{19}$$

$$= J^r(\pi^*) - J^r(\pi) + J^r(\pi) - \mathbb{E}_{\tau\sim\hat{\pi}}[g^r(\tau)] \tag{20}$$

$$\leq \epsilon\left(\frac{1}{\alpha_{f^r}} + 3\right)\mathcal{H}^2 + J^r(\pi) - \mathbb{E}_{\tau\sim\hat{\pi}}[g^r(\tau)]. \tag{21}$$

Next, for the second term in Equation 21:

$$J^r(\pi) - \mathbb{E}_{\tau\sim\hat{\pi}}[g^r(\tau)] \tag{22}$$

$$= \mathbb{E}_{\tau\sim\pi}[g^r(\tau)] - \mathbb{E}_{\tau\sim\hat{\pi}}[g^r(\tau)] \tag{23}$$

$$= \mathbb{E}_{\tau\sim\pi}[\sum_{t=1}^{\mathcal{H}}(r_t)] - \mathbb{E}_{\tau\sim\hat{\pi}}[\sum_{t=1}^{\mathcal{H}}(r_t)] \tag{24}$$

$$= \mathbb{E}_{\mathbf{s_1}}\sum_{t=1}^{\mathcal{H}}(P_t^r \cdot r_t) - \mathbb{E}_{\mathbf{s_1}}\sum_{t=1}^{\mathcal{H}}(\hat{P}_t^r \cdot r_t), \tag{25}$$

$$\tag{26}$$

where $P_t^r$ and $\hat{P}_t^r$ represent the probabilities of selecting the maximum-reward actions under policies derived from Equation 3 and Equation 13, respectively. Since rewards are binary and by the condition $P\{\hat{P}_i^r - P_i^r \geq \sigma_r, \forall i\} \geq 1 - \delta_r$, we have:

$$\mathbb{E}_{\mathbf{s_1}}\sum_{t=1}^{\mathcal{H}}(P_t^r \cdot r_t) - \mathbb{E}_{\mathbf{s_1}}\sum_{t=1}^{\mathcal{H}}(\hat{P}_t^r \cdot r_t) \tag{27}$$

$$= \mathbb{E}_{\mathbf{s_1}}\sum_{t=1}^{\mathcal{H}}[(P_t^r - \hat{P}_t^r)r_t] \tag{28}$$

$$\leq \mathbb{E}_{\mathbf{s_1}}\sum_{t=1}^{\mathcal{H}}(-\sigma_r) \cdot r_t \tag{29}$$

$$\leq -\mathcal{H}\sigma_r. \tag{30}$$

Substituting Equation 30 into Equation 21, we get:

$$\mathbb{E}_{\tau\sim\pi^*}[g^r(\tau)] - \mathbb{E}_{\tau\sim\hat{\pi}}[g^r(\tau)] \tag{31}$$

$$\leq \epsilon\left(\frac{1}{\alpha_{f^r}} + 3\right)\mathcal{H}^2 - \mathcal{H}\sigma_r. \tag{32}$$

$\square$

**Corollary E.5.** *If $\alpha_{f^r} > 0, \epsilon = 0$, and $f^r(s_1) = V^{r*}(s_1)$ for all initial states $s_1$, then $J^r(\pi^*) = J^r(\pi) = J^r(\hat{\pi})$ under $\hat{P}_i^r = P_i^r$ and $\sigma_r = 0$.*

*Remark* E.6. Because the corollary's conditions are stringent and seldom satisfied in practice, it is informative to compare against Lemma E.1. Relative to the standard SFT objective in Eq. 3, our framework (QP-SFT) achieves an additive improvement of $\mathcal{H}\sigma_r$, thereby tightening the objective and yielding superior policies compared with SFT baselines.

## F   MORE DISCUSSION

**Ablation study of hyperparameter sensitivity.** We present a comprehensive analysis of the impact of the hyperparameter $\alpha$ on the performance of two algorithms, QP-DPO and QP-SFT, across four continuous control environments in Figure 7. Experiments are conducted under both Medium and Expert data regimes, evaluating five values of $\alpha$ (0.001, 0.01, 0.1, 1, and 2).

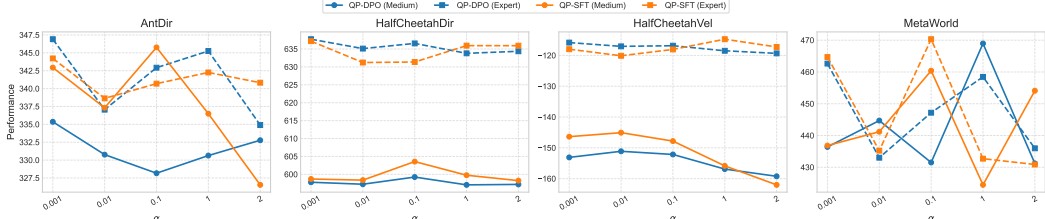

Figure 7: Performance of QP-DPO and QP-SFT algorithms across various tasks under different values of the hyperparameter $\alpha$.

Table 3: Performance of QP-DPO under various finetuning parameter configurations across different pretrained agent sizes in the MetaWorld benchmark. For certain configurations (e.g., Adaptor, Fullmodel), the number of trainable parameters increases with the size of the pretrained agent, while for others, the parameter count remains constant. The table reports both the finetuning parameter count and corresponding performance, providing a comprehensive comparison of algorithmic scalability and efficiency.

| | | | Medium | | |
|---|---|---|---|---|---|
| Size | 9.16KB | 9.16KB | 9.16KB | 9.16KB | 9.16KB |
| Prompt | $410.88 \pm 1.2$ | $419.78 \pm 1.3$ | $368.83 \pm 2.5$ | $591.68 \pm 5.0$ | $675.39 \pm 3.2$ |
| Size | 0.5M | 1MB | 2MB | 4MB | 6MB |
| Adaptor | $572.81 \pm 2.4$ | $460.43 \pm 3.3$ | $298.20 \pm 5.2$ | $593.51 \pm 3.5$ | $675.39 \pm 3.6$ |
| Size | 0.54M | 0.54M | 0.54M | 0.54M | 0.54M |
| Decorator | $449.68 \pm 20.2$ | $476.23 \pm 10.3$ | $277.14 \pm 24.4$ | $590.00 \pm 23.2$ | $672.58 \pm 10.2$ |
| Size | 6.61M | 12.66M | 49.32M | 97.52M | 145.73Mb |
| Fullmodel | $504.42 \pm 30.8$ | $694.87 \pm 20.3$ | $375.22 \pm 34.2$ | $650.33 \pm 23.5$ | $756.41 \pm 26.7$ |
| | | | Expert | | |
| Size | 9.16KB | 9.16KB | 9.16KB | 9.16KB | 9.16KB |
| Prompt | $410.88 \pm 1.1$ | $419.78 \pm 1.1$ | $368.83 \pm 1.7$ | $591.68 \pm 2.4$ | $675.39 \pm 3.0$ |
| Size | 0.5M | 1MB | 2MB | 4MB | 6MB |
| Adaptor | $505.46 \pm 2.0$ | $481.93 \pm 2.5$ | $349.31 \pm 4.2$ | $592.53 \pm 4.2$ | $675.39 \pm 3.0$ |
| Size | 0.54M | 0.54M | 0.54M | 0.54M | 0.54M |
| Decorator | $488.06 \pm 10.4$ | $475.68 \pm 14.5$ | $278.57 \pm 13.5$ | $588.15 \pm 14.7$ | $672.75 \pm 16.2$ |
| Size | 6.61M | 12.66M | 49.32M | 97.52M | 145.73Mb |
| Fullmodel | $504.49 \pm 20.5$ | $539.49 \pm 23.1$ | $393.38 \pm 25.2$ | $732.14 \pm 26.4$ | $691.17 \pm 28.3$ |

Our findings reveal several notable trends regarding the sensitivity of QP-DPO and QP-SFT to the hyperparameter $\alpha$. In the HalfCheetahDir and HalfCheetahVel environments, both algorithms demonstrate stable performance across the evaluated range of $\alpha$, indicating a relative insensitivity to this hyperparameter. Nevertheless, at higher values of $\alpha$ (e.g., $\alpha = 2$), QP-SFT exhibits a modest decline in performance, suggesting that excessive weighting may hinder effective policy optimization. In contrast, the AntDir environment displays greater sensitivity to $\alpha$. For QP-DPO, higher $\alpha$ values (0.1–1) correspond to improved performance, whereas QP-SFT experiences marked performance deterioration as $\alpha$ increases, particularly when utilizing expert datasets. Within the MetaWorld benchmark, QP-DPO achieves optimal performance at $\alpha = 1$ under the Medium data regime, while QP-SFT attains its best results at $\alpha = 0.1$. Both algorithms exhibit reduced stability at extreme values of $\alpha$ (i.e., $\alpha = 0.01$ and $\alpha = 2$), indicating that moderate values promote more stable and effective learning in complex, multi-task scenarios. Overall, moderate values of $\alpha$ (specifically, $\alpha = 0.1$ to $\alpha = 1$) yield more robust and consistent performance across the majority of environments and data regimes. In contrast, very small ($\alpha = 0.001$) or very large ($\alpha = 2$) values tend to introduce instability or lead to underfitting and overfitting effects.

**Ablation study on scaling pretrained agents.** As discussed in Section 6, we present a figure illustrating the performance implications of scaling pretrained agents. To provide a more comprehensive understanding, Table 3 offers a detailed quantitative analysis of these effects. The pretrained agents in our study are instantiated as Transformer-based agents, with model scale primarily controlled by varying the number of attention layers and attention heads. This approach allows us to systematically

Table 4: Runtime comparison across pretrained agent scales and finetuning algorithms.

| Runtime Comparison Across Pretrained Agent Scales (QP Algorithm) | | | | | |
|---|---|---|---|---|---|
| Pretrained agents scale | 5.83M | 14.61M | 65.32M | 129.52M | 193.73M |
| Prompt | 143.13 min | 236.29 min | 323.47 min | 592.63 min | 665.86 min |
| Adaptor | 162.16 min | 260.91 min | 367.86 min | 699.91 min | 745.77 min |
| Decorator | 158.43 min | 241.62 min | 336.59 min | 601.83 min | 666.47 min |
| Fullmodel | 147.39 min | 238.11 min | 327.34 min | 602.61 min | 670.85 min |
| Runtime Comparison Across Fine-Tuning Algorithms (Fixed Model Size) | | | | | |
| Algorithms | SFT | DPO | GRPO | PPO | CQL | QP |
| Prompt | 130.55 min | 136.30 min | 144.10 min | 143.80 min | 144.53 min | 143.13 min |
| Adaptor | 139.91 min | 144.31 min | 150.37 min | 151.51 min | 159.35 min | 162.16 min |
| Decorator | 141.36 min | 147.62 min | 150.29 min | 152.01 min | 157.46 min | 158.43 min |
| Fullmodel | 135.14 min | 138.35 min | 142.70 min | 142.29 min | 145.38 min | 147.39 min |

investigate the impact of model size on downstream performance. As shown in Table 3, increasing the size of the pretrained agents generally leads to improved performance across a variety of finetuning parameter configurations and finetuning dataset qualities. However, for Fullmodel finetuning, the substantial increase in trainable parameters introduced by larger architectures, when coupled with limited finetuning data, can result in performance saturation or even degradation. These results highlight the importance of balancing model capacity with the availability of high-quality supervision when designing scalable finetuning strategies. Careful consideration of this trade-off is crucial for maximizing performance gains while mitigating the risk of overfitting or inefficiency.

**Ablation study of computational cost.** We compare relative wall-clock time across algorithms and pretrained agent scales (Table 4). Because absolute runtime is hardware-dependent, our focus is on *relative* comparisons across algorithms and pretrained agent scales under a fixed setup. Two factors primarily drive runtime differences: (i) the size of the pretrained model, which governs forward/backward compute; and (ii) the complexity of the optimization objective, which varies modestly across finetuning strategies but remains of the same order of magnitude. Empirically, our method attains substantially higher performance at runtime comparable to strong baselines, indicating favorable efficiency at similar computational cost.

**Ablation study of main results.** To complement Table 1, Figure 8 presents a task-specific visualization that elucidates the relative performance across methodologies through normalized performance metrics. Error bars convey variability across random seeds, and per-task rankings make effect sizes more apparent than in tabular form. Across finetuning parameter configurations and environments, our proposed methods consistently deliver competitive performance, typically ranking within the top half of methods on most tasks.

**Ablation study on more complex tasks.** To further assess the scalability of our approach to larger models and more challenging environments, we conduct additional experiments using the LIBERO simulation benchmark (Liu et al., 2023a). LIBERO is designed to evaluate embodied agents on tasks requiring complex reasoning, spatial understanding, and long-horizon decision-making. It comprises four task suites:

- LIBERO-Spatial: Focuses on spatial relations and object positioning

- LIBERO-Object: Involves reasoning over diverse object categories

- LIBERO-Goal: Requires understanding and generalizing goal semantics

- LIBERO-Long (LIBERO-10): Comprises extended, sequential tasks that demand multi-step reasoning and planning

For these experiments, we adopt Dita (Hou et al., 2025), a transformer-based agent with 334M parameters (221M trainable). We finetune Dita for 40,000 steps using both standard SFT and our proposed QP-SFT approach. The model is trained on 8× NVIDIA RTX 4090 GPUs over three days. As shown in Table 5, QP-SFT consistently outperforms SFT across all task suites, with the largest gains observed in long-horizon settings—where structured policy improvement is most critical. These results demonstrate that QP-SFT scales effectively to larger models and more complex embodied environments.

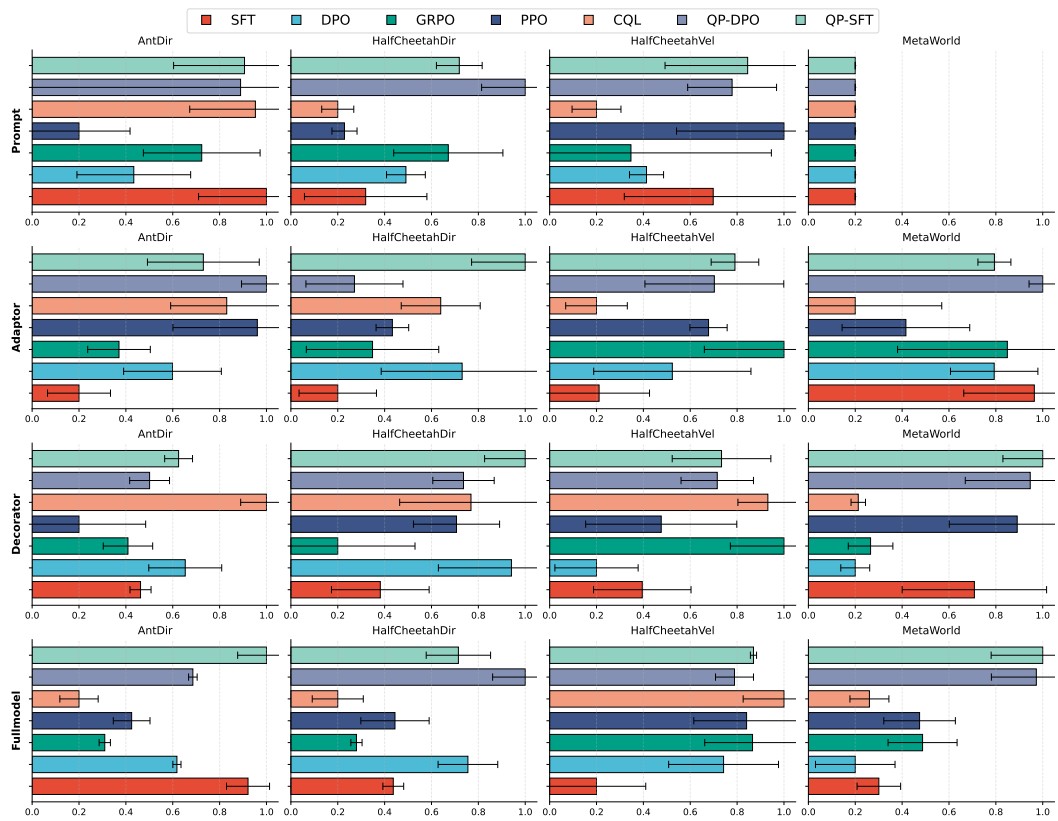

Figure 8: Normalized performance of different finetuning algorithms on meta-RL environments using expert-level offline datasets. Each method is evaluated with 50 finetuning trajectories, and results are averaged over three independent runs with different random seeds.

Table 5: Performance comparison of SFT and QP-SFT on the four LIBERO benchmark suites. QP-SFT yields consistent improvements, particularly on long-horizon tasks.

|  | Spatial | Object | Goal | Long |
|---|---|---|---|---|
| SFT | 84.2% | 96.3% | 85.4% | 63.8% |
| QP-SFT | 89.0% | 95.4% | 87.8% | 67.8% |

Table 6: Ablation comparing the original QP-SFT update rule and a max-sampling variant. While max-sampling may reduce value underestimation, it did not yield consistent performance gains in the limited-data regime.

|  | QP-SFT (original) | QP-SFT (max) |
|---|---|---|
| AntDir | $412.18 \pm 16.48$ | $405.22 \pm 20.30$ |
| HalfCheetahDir | $642.62 \pm 3.96$ | $644.37 \pm 6.2$ |
| HalfCheetahVel | $-119.79 \pm 0.26$ | $-120.74 \pm 2.32$ |
| MetaWorld | $553.36 \pm 35.35$ | $550.89 \pm 38.22$ |

**Ablation study on the $Q$-update rule.** We further investigate an alternative variant of the QP objective that samples multiple candidate actions and takes the maximum predicted value, instead of relying on a single predicted action. Specifically, we compare the original update formulation in

Equation 16 with the max-sampling variant:

$$\mathbb{E}_{(\mathbf{s},\mathbf{a},\mathbf{s}')\sim\mathcal{P}}\left|\left|\hat{Q}_m - Q_{\phi_i}(\mathbf{s},\mathbf{a})\right|\right|^2, \tag{33}$$

$$\text{where }\ \hat{Q}_m = r + \gamma \max_{\hat{\mathbf{a}}} Q_{\phi_i'}(\mathbf{s}',\hat{\mathbf{a}}), \tag{34}$$

where multiple actions are sampled for the maximization operator. This variant can in principle mitigate value underestimation, but it increases both variance and computational cost. In our limited-data setting, as shown in Table 6, it did not yield consistent improvements.

**Motivation for studying RFT in offline meta-RL.** Although RFT is primarily used in online RL settings, we highlight two key reasons why evaluating RFT in the offline meta-RL regime is both meaningful and practically important:

- **Practical motivation.** In many TA applications—such as robotics, embodied control, and safety-critical domains—online interaction is expensive, slow, or risky, making RLHF-style interactive finetuning infeasible. At the same time, TAs employ the same autoregressive Transformer backbone as LLMs, raising the natural question of whether RFT—highly effective in RLHF—can also improve adaptation from static trajectory datasets beyond what SFT provides. Prior work in these domains has focused almost exclusively on SFT (Zhao et al., 2022); the role of RFT in offline meta-RL remains largely unexplored. Our study fills this gap by evaluating whether RFT-style updates yield meaningful benefits under realistic data constraints.

- **Theoretical Justification.** RFT objectives such as PPO-, DPO-, and GRPO-style losses can be interpreted as regularized policy improvement schemes that do not intrinsically require online interaction, provided that a suitable advantage, value, or preference signal is available. In our QP formulation, we compute these signals through a learned $Q_\phi(s,a)$, yielding a KL-regularized policy improvement step aligned with recent work connecting preference-based updates and constrained policy optimization (Liu et al., 2024; Dai et al., 2025).

Thus, our goal is to test whether these RFT-style regularized updates can meaningfully improve offline policy adaptation of pretrained TAs compared to pure SFT, under realistic data constraints.

