# OpenReview forum: "The State of Reinforcement Finetuning for Transformer-based Agents"
_ICLR.cc/2026/Conference — ICLR 2026 Poster_

### Official Review · Reviewer_RqYc · 2025-10-20

**Soundness:** 1
**Presentation:** 1
**Contribution:** 1
**Rating:** 2
**Confidence:** 4

**Summary:**

The paper “The State of Reinforcement Finetuning for Transformer-based Generative Agents” aims to study how reinforcement finetuning (RFT) techniques, commonly used for aligning large language models such as through RLHF, can be applied to Transformer-based “generative agents.” It positions itself as a comprehensive evaluation of RFT methods in offline reinforcement learning settings, integrating variants such as reward modeling and policy-gradient finetuning. The authors claim that RFT can enhance sample efficiency and reasoning ability in pre-trained Transformer agents and present a benchmark framework comparing different RFT configurations. However, the paper’s objectives remain vague—it is unclear whether it introduces a new algorithm, proposes a benchmark, or provides theoretical insights—and the distinction between “Transformer-based generative agents” and standard LLMs or RL agents is not well defined.

**Strengths:**

The paper touches on a important topic. The motivation to unify reinforcement-based alignment with multi-task or offline RL adaptation reflects an original attempt to connect two active research directions. The authors make an effort to survey existing RFT approaches and to frame them within a consistent experimental setting, which could, in principle, contribute to clarifying the landscape of RL-based post-training methods. The writing is generally clear at a surface level, and the paper demonstrates awareness of recent developments in RLHF and RFT, attempting to position itself at the intersection of language modeling, reinforcement learning, and meta-adaptation. Overall, while the execution is weak, the underlying idea of systematically analyzing reinforcement-style finetuning for large Transformer-based agents shows conceptual ambition and topical relevance.

**Weaknesses:**

The main weakness of this paper lies in its lack of conceptual clarity and methodological grounding. It is unclear whether the paper’s goal is to propose a benchmark, a new algorithm, or an analytical study—and this ambiguity undermines its overall contribution. The term “Transformer-based generative agent” is used extensively without a precise definition or boundary relative to standard large language models, prompting confusion about what specific architecture or behavior distinguishes these agents from common RLHF-trained LLMs.

From a technical standpoint, the paper’s use of offline reinforcement learning tasks is poorly justified. RFT and RLHF are inherently online or preference-based alignment methods, not baselines for static offline RL datasets. The decision to evaluate RFT in this context suggests a fundamental misunderstanding of the underlying paradigms. Furthermore, the experimental setup is under-specified: there is no mention of which base model was used, what datasets were employed, or how the finetuning protocols were configured. Without this information, the reported results lack reproducibility and interpretability.

The evaluation design is also inadequate—the authors omit strong baselines such as CQL, IQL, or other modern offline RL methods, and they provide no ablations or sensitivity studies to validate their claims. The comparison across RFT variants remains superficial and largely descriptive, without clear analysis of why certain methods perform differently. Finally, the paper’s writing gives the impression of a survey-like overview rather than a focused, hypothesis-driven study; key references are discussed only at a surface level (e.g., RLHF and GRPO), and theoretical or empirical novelty is minimal.

**Questions:**

- Could the authors clarify the main objective of the paper — is it intended to serve as a benchmarking effort, a new RFT algorithm, or a conceptual analysis of reinforcement finetuning for Transformer-based agents? Clear positioning would help readers understand how to interpret the experimental results and contributions.

- How do the authors define a “Transformer-based generative agent” in contrast to standard LLMs fine-tuned with RLHF or RFT? Are these agents expected to operate in simulated environments, handle sequential decision-making, or simply produce text-conditioned reasoning? A more precise definition would make the scope of the study much clearer.

- Why were offline RL tasks chosen as the evaluation domain for RFT, which is traditionally an online or human-feedback-based training process? Are there theoretical or practical motivations for believing that RFT can meaningfully improve offline RL policy learning?

- What base model and pretraining setup were used for finetuning? Without information about the backbone architecture, scale, or initial performance, it is difficult to interpret whether the observed improvements come from RFT or model capacity.

- Could the authors provide stronger experimental baselines or ablation studies (e.g., comparing against standard offline RL algorithms such as CQL, IQL, or decision-transformer-based approaches) to substantiate the empirical claims?

---

> ### Author Response · Authors · 2025-11-21
>
> Thank you for your thoughtful review and your kind support. We have carefully addressed all of your comments below and thoroughly revised the manuscript, with all changes highlighted in red. We hope our responses satisfactorily resolve your concerns.
>
> > W1. The main weakness of this paper lies in its lack of conceptual clarity and methodological grounding. It is unclear whether the paper’s goal is to propose a benchmark, a new algorithm, or an analytical study—and this ambiguity undermines its overall contribution.
>
> >Q1. Could the authors clarify the main objective of the paper — is it intended to serve as a benchmarking effort, a new RFT algorithm, or a conceptual analysis of reinforcement finetuning for Transformer-based agents? Clear positioning would help readers understand how to interpret the experimental results and contributions.
>
> A1. We thank the reviewer for this important concern. Our intention is not to present a pure benchmark, but to deliver a **conceptual and methodological study** of RFT for Transformer-based decision agents, with QP as a minimal but principled extension.
>
> **Novel empirical question and cross-domain investigation:** RFT has been developed primarily for LRMs, yet its **transferability to Transformer-based generative agents (TGAs)**—which act in *sequential decision-making* environments rather than text domains—has not been systematically examined. Our work is, to our knowledge, the **first systematic study** of whether and when RFT-style updates that succeed in LRMs also benefit TGAs in meta-RL settings, which is a necessary step toward general-purpose agents.
>
> **Methodological enhancement:** QP integrates Q-value guidance directly into *policy-level finetuning objectives* (Eqs. 13–14), explicitly bridging **supervised imitation (SFT/DPO)** and **reinforcement-based policy improvement** in the few-shot offline regime. This design is motivated by the observed complementary strengths of SFT and RFT in our experiments. Moreover, we provide **theoretical support**—including a bound on the deviation from SFT (Theorem E.4 and Corollary E.5).
>
> **Breadth and interpretive depth:** Beyond proposing QP, we conduct a **comprehensive empirical analysis** across (i) four finetuning parameter configurations, (ii) seven finetuning algorithms, (iii) two meta-RL benchmarks, and (iv) ablations on data quality, data quantity, reward sparsity, and model scale. This reveals previously unreported patterns in offline meta-RL settings, such as RFT methods outperforming SFT under expert data but degrading on medium-quality data.
>
> **Clarifying Observed Behavior:** While the focus of this paper is empirical, we **do not merely report numbers**. Throughout Sections 5 and 6, we provide **reasoned analyses and ablation studies** to explain key observations, such as the performance tradeoffs between SFT and RFT under different data regimes, and the role of Q-function guidance in stabilizing policy updates.
>
> In summary, our contribution lies in (i) extending RFT analysis to a new class of Transformer-based decision agents, (ii) introducing a theoretically grounded Q-guided hybrid finetuning objective, and (iii) providing the first systematic empirical characterization of RFT in this setting for practical deployment. Thus, the paper’s contribution is best characterized as a **conceptual and empirical investigation [1,2,3]** that seeks to identify *when* and *why* RFT can improve Transformer-based policy adaptation.
>
> We will make this framing explicit in the title and introduction to avoid ambiguity.
>
> [1] On the Effectiveness of Fine-tuning Versus Meta-reinforcement Learning, NeurIPS 2022.
>
> [2] The State of Sparse Training in Deep Reinforcement Learning, ICML 2022.
>
> [3] Revisiting Plasticity in Visual Reinforcement Learning: Data, Modules and Training Stages, ICLR 2024.

---

> > ### Author Response · Authors · 2025-11-21
> >
> > >W1. The term “Transformer-based generative agent” is used extensively without a precise definition or boundary relative to standard large language models, prompting confusion about what specific architecture or behavior distinguishes these agents from common RLHF-trained LLMs.
> >
> > > Q2. How do the authors define a “Transformer-based generative agent” in contrast to standard LLMs fine-tuned with RLHF or RFT? Are these agents expected to operate in simulated environments, handle sequential decision-making, or simply produce text-conditioned reasoning? A more precise definition would make the scope of the study much clearer.
> >
> > A2. Thanks for the valuable suggestion.
> >
> > In our work, a **Transformer-based generative agent (TGA)** is defined as:
> >
> > “A Transformer-based model that generates action sequences conditioned on past state–action–reward histories rather than textual tokens, and that can generalize to unseen tasks through few-shot trajectory conditioning.”
> >
> >
> > Conceptually, our use of “generative agents” follows the **trajectory-modeling perspective in control**: these agents use **autoregressive generative models** (e.g., Transformer decoders) to produce action sequences given states and returns, analogous to next-token prediction in LLMs. This is aligned with formulations such as **Decision Transformer [1]** and **Q-Transformer [2]**, where policies are treated as generative sequence models over trajectories.
> >
> > They share the *same architectural backbone* as LLMs but differ in **output semantics** and **learning objectives**:
> >
> > - **LLMs with RLHF/RFT:** model $p(y_t | y_{<t})$ over text tokens, optimizing for human-preference rewards.
> > - **TGAs:** model $p(a_t | s_{\le t}, a_{<t}, r_{\le t})$ over actions conditioned on trajectories and are optimized for **task rewards or return-to-go**, operating in **simulated control environments** and evaluated by task-specific reward metrics rather than preference alignment.
> >
> > We will add this formal definition and clarify in Section 2.1
> >
> > [1] Decision Transformer: Reinforcement Learning via Sequence Modeling
> >
> > [2] Q-Transformer: Scalable Offline Reinforcement Learning via Autoregressive Q-Functions

---

> > > ### Author Response · Authors · 2025-11-21
> > >
> > > >W2. From a technical standpoint, the paper’s use of offline reinforcement learning tasks is poorly justified. RFT and RLHF are inherently online or preference-based alignment methods, not baselines for static offline RL datasets. The decision to evaluate RFT in this context suggests a fundamental misunderstanding of the underlying paradigms.
> > >
> > > > Q3. Why were offline RL tasks chosen as the evaluation domain for RFT, which is traditionally an online or human-feedback-based training process? Are there theoretical or practical motivations for believing that RFT can meaningfully improve offline RL policy learning?
> > >
> > > A3. We appreciate this question and confirm that the use of **offline RL tasks** is a deliberate design choice rather than a misunderstanding of the RFT paradigm.
> > >
> > > Our motivation is twofold:
> > >
> > > 1. **Practical motivation:**
> > > In many TGA applications—e.g., robotics and embodied agents—**online interaction is expensive, slow, or safety-critical**, making RLHF-style interactive finetuning impractical. At the same time, TGAs share the same autoregressive Transformer backbone as LLMs, so it is natural to ask whether the **policy-improvement dynamics of RFT** (successful in RLHF) can also improve **adaptation from static trajectory datasets**, beyond standard SFT. Prior work in these domains has focused almost exclusively on SFT-style finetuning; the role of RFT in this offline meta-RL regime has remained largely unexplored. Thus, our work aims to fill precisely this gap, giving insights that whether RFT could improve offline meta-RL finetuning.
> > > 2. **Theoretical Justification:**
> > > RFT objectives such as PPO- and GRPO-style losses can be written as **regularized policy improvement** schemes that do *not inherently require* online interaction, provided that a suitable **advantage, value, or preference signal** is available. In our QP formulation, we extend this view by using a learned $Q_\phi(s,a)$ which yields a **KL-regularized policy improvement step** in line with recent works connecting preference-based updates and constrained policy optimization [1,2].
> > >
> > > Thus, our goal is to test whether these **RFT-style regularized updates** can meaningfully improve **offline policy adaptation** of pretrained TGAs compared to pure SFT, under realistic data constraints. Empirically, we find that—even though RFT is not traditionally tailored for strictly offline adaptation—it can achieve **surprisingly strong performance** in this setting. This motivates our **systematic investigation** of *when* and *under what conditions* such behavior occurs, which we view as the central empirical contribution of the paper.
> > >
> > > We will make these theoretical and practical motivations explicit in the revised manuscript.
> > >
> > > [1] Provably Mitigating Overoptimization in RLHF: Your SFT Loss is Implicitly an Adversarial Regularizer, NeurIPS 2024
> > >
> > > [2] Mitigating reward over-optimization in rlhf via behavior-supported regularization, ICLR 2025.

---

> > > > ### Author Response · Authors · 2025-11-21
> > > >
> > > > > W2. Furthermore, the experimental setup is under-specified: there is no mention of which base model was used, what datasets were employed, or how the finetuning protocols were configured. Without this information, the reported results lack reproducibility and interpretability.
> > > >
> > > > > Q4. What base model and pretraining setup were used for finetuning? Without information about the backbone architecture, scale, or initial performance, it is difficult to interpret whether the observed improvements come from RFT or model capacity.
> > > >
> > > > A4. We apologize for the lack of explicit specification and will provide full details in the revision.
> > > >
> > > > **Backbone architecture and scale:** As described around L106–118, we use **Prompt-DT** as the base model, built on a minGPT-style Transformer decoder. The number of layers, hidden dimension, and attention heads vary with model scale, ranging from (3  layers, 128 dims, 1 head) to (48 layers, 256 dims, 16 heads), corresponding to **5M–193M parameters** (see also in Appendix C/E).
> > > >
> > > > **Pretraining data and tasks:** Pretraining datasets are generated using **Soft Actor–Critic (SAC)** experts on standard **MuJoCo locomotion tasks** (AntDir, CheetahDir, CheetahVel) and **MetaWorld manipulation tasks** (Appendix B).
> > > >
> > > > - For **MuJoCo**, the pretraining dataset contains **4,500 trajectories**, while finetuning uses **50 trajectories**, i.e., roughly **1.1%** of the pretraining data.
> > > > - For **MetaWorld**, the pretraining dataset contains **45,000 trajectories**, and again **50 trajectories** are used for finetuning, i.e., about **0.1%** of the pretraining data.
> > > >
> > > > **Initial performance and finetuning protocols:** The **Zero-shot** column in Table 1 reports the performance of the **pretrained backbone before any finetuning**, providing a direct reference for the gains achieved by each method. All finetuning methods start from **identical pretrained checkpoints**, so performance differences arise from the finetuning procedure rather than model capacity. For each method, we report the **best checkpoint** on the held-out meta-test tasks, following standard meta-RL practice to account for differing convergence dynamics.
> > > >
> > > > We will emphasize these details to ensure the setup is fully transparent.
> > > >
> > > > > W3. The evaluation design is also inadequate—the authors omit strong baselines such as CQL, IQL, or other modern offline RL methods, and they provide no ablations or sensitivity studies to validate their claims. The comparison across RFT variants remains superficial and largely descriptive, without clear analysis of why certain methods perform differently.
> > > >
> > > > > Q5. Could the authors provide stronger experimental baselines or ablation studies (e.g., comparing against standard offline RL algorithms such as CQL, IQL, or decision-transformer-based approaches) to substantiate the empirical claims?
> > > >
> > > > A5. We appreciate this concern and agree that baseline choice and analysis need to be clearly justified.
> > > >
> > > > Our current experimental suite already includes **CQL** as a strong offline RL baseline, alongside **PPO**, **SFT**, and two representative **RFT/RLHF-style methods (DPO, GRPO)**. Given the breadth of factors we study—four finetuning parameter configurations, multiple data regimes (Medium vs. Expert, different trajectory numbers), reward sparsity, and model scale—it is practically difficult to include *all* prominent offline RL algorithms (e.g., IQL, additional DT variants) without substantially reducing coverage along these other axes. Our design goal was therefore to focus on a **representative set of widely used methods** from three families:
> > > >
> > > > - **Supervised imitation:** SFT
> > > > - **RLHF / RFT-style:** DPO, GRPO
> > > > - **Classic RL / offline RL:** PPO, CQL
> > > >
> > > > and to examine how these families behave under a **shared TGA backbone and identical data conditions.** Our primary objective is to **systematically compare SFT vs. RFT (and QP)** across finetuning parameter configurations and data regimes, so as to **provide guidance for method selection** in similar scenarios.
> > > >
> > > > That said, we agree that the **relative behavior of these methods** merits clearer interpretation. In the revision, we will (i) explicitly justify our finetuning algorithm selection, and (ii) expand the analysis to discuss key patterns and counterintuitive results—such as PPO’s behavior in sparse vs. dense reward settings (Answer 11 to Reviewer **EYuE**) and the identical MetaWorld performance under prompt tuning  (Answer 8 to Reviewer **EYuE**) —to provide more insight into *why* different RFT and SFT variants behave as observed.

---

> > > > > ### Author Response · Authors · 2025-11-21
> > > > >
> > > > > > W3. Finally, the paper’s writing gives the impression of a survey-like overview rather than a focused, hypothesis-driven study; key references are discussed only at a surface level (e.g., RLHF and GRPO), and theoretical or empirical novelty is minimal.
> > > > >
> > > > > A6. We appreciate this comment and understand the concern about the paper appearing survey-like rather than hypothesis-driven.
> > > > >
> > > > > Our intent is **not** to write a survey, but to conduct a **focused empirical study** of RFT for Transformer-based generative agents. Concretely, the paper is organized around the following guiding questions:
> > > > >
> > > > > - *When and how does RFT outperform (or underperform) SFT in few-shot meta-RL adaptation?*
> > > > > - *How do data quality, data quantity, reward sparsity, model scale, and parameterization (prompt / adaptor / decorator / full) modulate this relationship?*
> > > > > - *Can a simple Q-guided hybrid (QP) improve over both pure SFT and pure RL-style finetuning in this regime?*
> > > > >
> > > > > To answer these questions in a way that yields **generalizable takeaways**, we deliberately:
> > > > >
> > > > > - Focus on a **representative set of widely used algorithms** (SFT, DPO, GRPO, PPO, CQL, and QP variants), rather than exhaustively covering all state-of-the-art methods. These are precisely the **methods practitioners are most likely to try first**.
> > > > > - Design a **large, factorial experimental grid** (seven algorithms × four environments × four parameter configurations, further crossed with data quality, trajectory count, reward sparsity, and model scale) so that conclusions are based on **systematic patterns** rather than single-case anecdotes.
> > > > >
> > > > > In addition, we provide **theoretical support** for the QP, including bounds that relate its behavior to SFT, with the goal of offering insight into *how* and *why* such value-guided finetuning can improve performance.
> > > > >
> > > > > In the revision, we will make these guiding questions and the theoretical role of QP more explicitly, to clearly distinguish the work from a survey and to highlight its empirical and conceptual contributions.

---

### Official Review · Reviewer_EYuE · 2025-10-30

**Soundness:** 3
**Presentation:** 2
**Contribution:** 3
**Rating:** 4
**Confidence:** 4

**Summary:**

This work attempts to provide an earnest and comprehensive comparison view on the state of reinforcement fine-tuning (RFT) compared to supervised fine-tuning (SFT) of RL agents. The authors identify several RFT and SFT methods commonly used in the literature for RFT used for LLMs and apply them to RL agents on different environments (Metaworld, Mujoco) in the few-shot scenario to sparse and dense reward settings. Moreover based on the current observation of methods the authors propose a new method that combines RFT and SFT. Across a wide range of experiments the authors demonstrate the benefits of their newly proposed method, which is usually among the best performing methods. Finally, the authors try to provide useful takeaways beneficial for practitioners.

**Strengths:**

- The paper attempts to provide an earnest and comprehensive comparison of SFT vs RFT on RL tasks.
- Plenty of experiments for different fine-tuning methods (SFT vs RFT)
- The proposed method seems to generally perform pretty well on the different settings.

**Weaknesses:**

**Inconsistent results and takeaways**

I like the approach of the authors to try to break down plenty of results into key takeaways, however I found multiple contradictions to the key takeaways, some of which are listed below. This is the main factor why I am currently leaning towards rejection at the moment, if they can be resolved I'd consider increasing my rating.

The paragraph on line 312 states the superiority of QP algorithms and their robustness and broad applicability. Taking a closer look at Table 1, however, the gap to competitors is not large and there is usually a competitor that lies within one standard deviation of QP (e.g. DPO vs QP-SFT for Adapters and Decorators). This means that if a test for statistical significance was performed, the result would be that there is no statistically significant difference. Therefore I recommend to tone down the claiming on superiority and robustness of QP.

In line 368 states that increasing the number of trajectories consistently improves performance for all methods, however looking at e.g. PPO this is not the case. Similarly performance on DPO for expert is sometimes worse with 100 trajectories than with 50 for expert data. Furthermore, Figure 1 is utterly confusing to me, why is there a distinction in dataset quality for different methods? Does that mean the different methods were trained on different fine-tuning data? Or does it just show the same scores as Figure 2, but normalized? If the latter is the case, please correct the caption because at the moment it states "Comparison of finetuning dataset quality" which led to my confusion. Furthermore, if I interpreted it correctly and they essentially show the same information, one of those figures could be moved to appendix to accomodate space for further interpretation/results.

Another example are takeaways in line 372: "Supervised approaches yield stable, imitation-driven performance—particularly
when data quality is mixed", however SFT is one of the worst methods on HalfCheetah (Medium). Yet another example: "(ii) regardless of the algorithmic family, more finetuning trajectories monotonically enhance performance", however CQL on Metaworld (Expert) gets worse with more trajectories.

**Novelty**

One important aspect of the contributions of this work is the investigation of the data quality for fine-tuning, however this has been investigated in prior work already [1]. In particular, this prior work introduces metrics quantifying trajectory quality and state-action coverage and their influence on performance for several offline RL algorithms. It would be helpful to clarify the distinction to this work.

Furthermore, [2] also provides a comprehensive comparison between different parameter-efficient fine-tuning approaches for single-task fine-tuning on Metaworld and DMControl, including a wider variety of PEFT methods than this work. Though they only investigate SFT and no RL fine-tuning. It would be helpful to outline the difference to those works concretely.

[1] A dataset perspective on offline reinforcement learning, Schweighofer et al., CoLLAs 2022

[2] Learning to Modulate pre-trained Models in RL, Schmied et al., NeurIPS 2023

**Conceptual framing**

The authors mention several times throughout the paper that they investigate a metaRL setup, however to me it seems it is more like an actual transfer learning/ multitask fine-tuning setup as in [1]. There are also no conventional metaRL algorithms compared which makes me wonder whether the "metaRL" framing is the correct terminology. Furthermore the terminology of "generative agents" is rather confusing to me, as this terminology is usually employed for LLM-based agentic frameworks [2]. Also the focus on "transformer-based" is not necessary as all the tested algorithms are agnostic to the architecture.

[1] Learning to Modulate pre-trained Models in RL, Schmied et al., NeurIPS 2023

[2] Generative Agents: Interactive Simulacra of Human Behavior, Park et al., UIST 2023



**Choice of methods**

It might be interesting looking into [1] for a LoRA-variant that has been shown to significantly improve upon LoRA based single-task finetuning on the MetaWorld tasks in [2]. This could potentially affect the ranking of the methods in the experiments.

[1] Parameter Efficient Fine-tuning via Explained Variance Adaptation, Paischer et al., ENLSP workshop at NeurIPS 2024

[2] Learning to Modulate pre-trained Models in RL, Schmied et al., NeurIPS 2023

**Questions:**

- Eq 15: why is the expectation only over (s,a) tuples? why can we not just as in regular double Q-learning use (s,a,s’) tuples for training?
 - Eq 16: would it make a difference if you sampled multiple actions from the policy and take the max instead of only using $\hat{a}$? I know $\pi_\theta$ already maximizes the q-value, but it might help wich counteracting approximation errors.
 - Line 261 mentions that the fine-tuning dataset is much smaller than the pretraining dataset, it would be helpful to provide actual numbers here. While Table 1 mentions 50 finetuning trajectories, this should be made more explicit in the text and not just in the caption.
 - Table 1: Why is the performance on MetaWorld for Prompt tuning equal across all methods?
 - Line 320 mentions that PPO and CQL lack inductive priors, what does that mean?
 - Line 323 mentions that switching from Adapters to Decorator for PPO yields 3% improvement, however the average score of Adapters for PPO is larger than the one of Decorator, am I missing something?
 - The final reached reward for the sparse setting (Figure 3) is around the same (sometimes even higher) than for the dense setting, why? What is the delay in the reward? For the same number of update steps I would expect massive differences if there was a substantial delay in the reward as shown in [1], which does not seem to be the case for e.g. PPO or CQL.
 - Any intuition as to why PPO trained on expert data is better in sparse reward settings than in dense reward settings?

 [1] RUDDER: Return Decomposition for Delayed Rewards, Arjona-Medina et al., NeurIPS 2019

---

> ### Author Response · Authors · 2025-11-21
>
> Thank you for your thoughtful review and your kind support. We have carefully addressed all of your comments below and thoroughly revised the manuscript, with all changes highlighted in red. We hope our responses satisfactorily resolve your concerns.
>
> > W1. Inconsistent results and takeaways. The paragraph on line 312 states the superiority of QP algorithms ... however CQL on Metaworld (Expert) gets worse with more trajectories.
>
> A1. We sincerely thank the reviewer for their careful and insightful review. We acknowledge that some of our descriptions may have overstated consistency, and we will revise the text accordingly to better reflect the nuances of the results.
>
> 1. **On QP “superiority and robustness” (L312).**
>
>     We agree that while QP methods (QP-DPO, QP-SFT) show the highest *mean* performance across environments (Table 1), the differences from strong baselines like DPO and GRPO are often within one standard deviation. We will **rephrase our claims** to emphasize the consistent *improvement trends* observed in most cases, rather than overstating the magnitude of the differences. Furthermore, as we did not cover all possible environments and configurations, we focus on **mean performance** to identify the most statistically robust methods. This will be clarified in the revision.
>
> 2. **On monotonic trajectory improvements (L368).**
>
>     The reviewer is correct that some individual runs (e.g., PPO or DPO) show non-monotonic behavior due to stochasticity and finite sample variance. Our intention was to convey that **the general trend across methods** shows a positive correlation between trajectory count and performance (see Fig. 2, averaged across settings). We will clarify that this is a *dominant but not universal* trend, and we will soften the language to reflect this variability.
>
> 3. **On Figure 1 caption and confusion.**
>
>     In **Figure 1**, we focus on the effect of *fine-tuning dataset quality* under a fixed trajectory budget of 50. We distinguish two quality levels, **medium** and **expert**, determined by how many random/low-quality trajectories are included when constructing the fine-tuning dataset (Detailed in Appendix B). In principle, this setting spans seven algorithms × four environments × four finetuning parameter configurations × two dataset qualities. To make the visualization interpretable, we first average over the four finetuning configurations, effectively reducing the display to **seven algorithms × four environments × two qualities**. We then plot two panels, one per quality level, where each panel shows the performance of all seven algorithms across the four meta-RL tasks. Thus, the caption “Comparison of fine-tuning dataset quality” is intended to convey: *given different fine-tuning dataset qualities (medium vs. expert), how do the methods compare across the four tasks, under a fixed trajectory budget of 50?*
>
>     In **Figure 2**, we jointly analyze the impact of the *number of fine-tuning trajectories* and *fine-tuning dataset quality*. Before aggregation, this setting involves seven algorithms × four environments × four finetuning parameter configurations × two qualities × three trajectory counts. As in Figure 1, we average over the four finetuning parameter configurations for clarity. Each panel in Figure 2 then fixes one environment and one dataset quality, and plots **seven algorithms × three trajectory counts**, so that the reader can see how performance scales with additional trajectories for each method. Conceptually, **Figure 1 can be viewed as a clearer, fine-grained slice of Figure 2** at the specific trajectory count of 50, designed to isolate and better visualize the effect of dataset quality alone, while Figure 2 provides the more comprehensive but denser picture of how both quality and trajectory number interact.
>
> 4. **On SFT and CQL performance exceptions (L372).**
>
>     We apologize for the confusion in the original text. SFT achieves top-tier performance in most tasks, but its performance can degrade when the number of tuning trajectories is small. This suggests that SFT requires a certain scale of data (e.g., more than 50 trajectories) to perform optimally. We will revise the takeaway to state that **SFT** typically performs well when **trajectory scale** exceeds a certain threshold.
>
>     - “Supervised approaches yield stable, imitation-driven performance—particularly when data quality is mixed and when the number of trajectories exceeds a certain scale.”
>     - “regardless of the algorithmic family, more finetuning trajectories typically enhance performance;”
>
>     These adjustments will ensure that the takeaways more accurately align with the data presented and provide a more rigorous interpretation of the results.

---

> > ### Author Response · Authors · 2025-11-21
> >
> > > W2. Novelty. One important aspect of the contributions of this work is ... outline the difference to those works concretely.
> >
> > A2. We thank the reviewer for highlighting related studies and agree that positioning relative to prior work can be clearer.
> >
> > 1. **Distinction from Schweighofer et al. (2022)**
> >
> >     Schweighofer et al. (2022) focus on **dataset quality** metrics in offline RL, quantifying trajectory return and state-action coverage of behavioral policies to correlate these features with the performance of classical offline RL algorithms trained from scratch. Our study, by contrast, explores how **data quality** influences **few-shot finetuning** in **meta-RL environments** with pre-trained models. Specifically, we examine how the proficiency of **finetuning trajectories** (Medium vs. Expert), along with **dataset size** and **reward sparsity**, affects the performance of various finetuning algorithms (SFT, DPO, GRPO, PPO, CQL, and QP) across multiple parameter configurations. While Schweighofer et al. focus on offline RL trained on fixed replay buffers, our work investigates the interplay between **pre-trained generative agents** and the **quality/quantity of adaptation data** in **high-dimensional, meta-RL settings**. Our study is thus complementary, focusing on **task-conditioned adaptation** rather than **global state-action coverage**, and asking how **adaptation data** shapes the success of RFT vs. SFT in few-shot regimes.
> >
> > 2. **Distinction from Schmied et al. (2023)**
> >
> >     Schmied et al. (2023) introduce the **Learning-to-Modulate (L2M)** framework, which focuses on **parameter-efficient fine-tuning (PEFT)** and **prompt-based tuning (PBT)** for continual-task RL in environments like MetaWorld, DMControl and Continual-World. Their work aims to mitigate catastrophic forgetting while adapting pre-trained models to new tasks. In contrast, our work investigates **reinforcement finetuning (RFT)** across both **full-model and parameter-efficient** strategies for Transformer-based generative agents, with a specific focus on **meta-RL tasks**. While both works explore fine-tuning, **L2M** is optimized for **sequential-task learning to address catastrophic forgetting** in well-defined environments, whereas our study examines how different finetuning algorithms, including **RFT-based methods** like PPO and QP-DPO, handle task adaptation in **few-shot, sparse-reward** settings. Thus, while both works address fine-tuning, **L2M** focuses on mitigating forgetting in continuous tasks, whereas **RFT** targets **generalization and adaptation** in more diverse RL settings, particularly for **large-scale generative agents**. The two are complementary in addressing different aspects of task adaptation.
> >
> > [1] A dataset perspective on offline reinforcement learning, Schweighofer et al., CoLLAs 2022
> >
> > [2] Learning to Modulate pre-trained Models in RL, Schmied et al., NeurIPS 2023

---

> > > ### Author Response · Authors · 2025-11-21
> > >
> > > > W3. Conceptual framing. The authors mention several times throughout the paper that they investigate a metaRL setup, however to me it seems it is more like an actual transfer learning/ multitask fine-tuning setup as in [1]. There are also no conventional metaRL algorithms compared which makes me wonder whether the "metaRL" framing is the correct terminology. Furthermore the terminology of "generative agents" is rather confusing to me, as this terminology is usually employed for LLM-based agentic frameworks [2]. Also the focus on "transformer-based" is not necessary as all the tested algorithms are agnostic to the architecture. [1] Learning to Modulate pre-trained Models in RL, Schmied et al., NeurIPS 2023 [2] Generative Agents: Interactive Simulacra of Human Behavior, Park et al., UIST 2023
> > >
> > > A3. We thank the reviewer for these thoughtful comments on terminology and framing. We agree that greater clarity will improve the paper, and we will revise the text accordingly.
> > >
> > > - **On “meta-RL” vs. “multi-task transfer.”**
> > >
> > >     Our experiments follow the meta-adaptation paradigm—training on a distribution of tasks and evaluating on unseen tasks—which is consistent with standard offline meta-RL formulations [1, 2]. We do not, however, adopt conventional meta-RL algorithms because our setting assumes a pretrained agent obtained from large-scale training, rather than a meta-policy optimized across tasks. Instead, our focus is to systematically compare *RFT* and *SFT* on downstream tasks in this pretrained-agent regime, which differs in scope and objectives from classical meta-RL.
> > >
> > > - **On the term “generative agents.”**
> > >
> > >     Our use of “generative agents” follows the trajectory-modeling perspective in decision-making: these agents employ **autoregressive generative models** (e.g., Transformer decoders) to produce sequences of actions conditioned on states and returns, analogous to next-token prediction in LLMs. This aligns with formulations such as **Decision Transformer**  [3] and **Q-Transformer [4]** , where policies are treated as generative sequence models over trajectories.
> > >
> > > - **On the emphasis on “Transformer-based.”**
> > >
> > >     While the proposed finetuning algorithms are architecturally agnostic, we focus on Transformers because they currently form the dominant backbone for **scalable, general-purpose decision models** and share training dynamics with LLMs—making RFT particularly relevant. We agree that other architectures may exhibit different behaviors and will note this as an avenue for future research.
> > >
> > > - **Distinction from language-based agents.**
> > >
> > >     Unlike LLM-based agent frameworks, where the model *acts through language*, the TGAs in our study operate in continuous-control environments: they take structured state inputs and generate **action sequences**, not natural language. We will explicitly define this term to distinguish it from the “LLM-based agentic frameworks” of Park et al. (2023), which focus on language-as-action interactions.
> > >
> > >
> > > [1] Offline Meta-Reinforcement Learning with Advantage Weighting
> > >
> > > [2] Prompting Decision Transformer for Few-Shot Policy Generalization
> > >
> > > [3] Decision Transformer: Reinforcement Learning via Sequence Modeling
> > >
> > > [4] Q-Transformer: Scalable Offline Reinforcement Learning via Autoregressive Q-Functions

---

> > > > ### Author Response · Authors · 2025-11-21
> > > >
> > > > > W4. Choice of methods. It might be interesting looking into [1] for a LoRA-variant that has been shown to significantly improve upon LoRA based single-task finetuning on the MetaWorld tasks in [2]. This could potentially affect the ranking of the methods in the experiments.
> > > > [1] Parameter Efficient Fine-tuning via Explained Variance Adaptation, Paischer et al., ENLSP workshop at NeurIPS 2024
> > > > [2] Learning to Modulate pre-trained Models in RL, Schmied et al., NeurIPS 2023
> > > >
> > > > A4. We thank the reviewer for pointing us to this line of work. Paischer et al. [1] propose an Explained Variance Adaptation (EVA) LoRA variant with data-driven initialization and adaptive rank, which indeed shows strong gains for parameter-efficient finetuning on MetaWorld in [2]. We agree that incorporating EVA-style modules into our **Adaptor** configuration (where LoRA is inserted into Transformer MLP layers) could further improve absolute performance and may slightly affect the ranking *within* specific finetuning parameter configurations.
> > > >
> > > > However, our primary objective in this paper is to **systematically compare SFT vs. RFT** across **multiple finetuning parameter configurations and data regimes**, rather than to optimize any single PEFT technique. Most of our key conclusions and ablations are reported **averaged over prompt, adaptor, decorator, and full-model configurations**, which we select because they are widely used and representative in the finetuning literature.
> > > >
> > > > We agree that extending our study to include EVA and other advanced PEFT methods is a valuable direction, and we will explicitly mention this as future work when systematically discussing the influence of finetuning parameterization and possible initialization on the finetuning behavior.
> > > >
> > > > > Q1. Eq 15: why is the expectation only over (s,a) tuples? why can we not just as in regular double Q-learning use (s,a,s’) tuples for training?
> > > >
> > > > A5. Thank you for pointing this out. This is indeed a typo error. It should be
> > > >
> > > > $E_{(s,a,s’) \sim P} ||\hat{Q}\_m - Q\_{\phi_i}(s,a)||^2 $
> > > >
> > > > where $\hat{Q}\_m = r + \gamma \min_{i=1,2} Q_{\phi’_i}(s’, \hat{a})$.
> > > >
> > > > We will correct this in the revision.
> > > >
> > > > > Q2. Eq 16: would it make a difference if you sampled multiple actions from the policy and take the max instead of only using $\hat{a}$? I know $\pi_{\theta}$ already maximizes the q-value, but it might help wich counteracting approximation errors.
> > > >
> > > > A6. Thank you for this suggestion. We investigated it with a small ablation.
> > > > Concretely, we compare the original QP-SFT (original) against a variant **QP-SFT (max)** that samples multiple actions and takes the maximum Q-value among them. Using the full-model setting across four environments, we obtain:
> > > >
> > > > |  | QP-SFT (original) | QP-SFT(max) |
> > > > | --- | --- | --- |
> > > > | AntDir | $412.18 \pm 16.48$ | $408.22 \pm 20.30$ |
> > > > | HalfCheetahDir | $642.62 \pm 3.96$ | $644.37 \pm 6.20$ |
> > > > | HalfCheetahVel | $-119.79 \pm 0.26$ | $-120.74 \pm 2.32$ |
> > > > | MetaWorld | $553.36 \pm 35.35$ | $550.89 \pm 38.22$ |
> > > >
> > > > Sampling multiple actions and taking the max can, in principle, mitigate underestimation, but it also increases variance and computational cost. In our experiments, it did **not** yield consistent improvements under the limited-data regime. We will mention this trade-off and our empirical findings in the revision.
> > > >
> > > > > Q3. Line 261 mentions that the fine-tuning dataset is much smaller than the pretraining dataset, it would be helpful to provide actual numbers here. While Table 1 mentions 50 finetuning trajectories, this should be made more explicit in the text and not just in the caption.
> > > >
> > > > A7. Thank you for the suggestion.
> > > >
> > > > For **MuJoCo**, the pretraining dataset contains **4,500 trajectories**, while finetuning uses **50 trajectories**, i.e., roughly **1.1%** of the pretraining data.
> > > >
> > > > For **MetaWorld**, the pretraining dataset contains **45,000 trajectories**, and again **50 trajectories** are used for finetuning, i.e., about **0.1%** of the pretraining data.
> > > >
> > > > We will add these into the text.
> > > >
> > > > > Q4. Table 1: Why is the performance on MetaWorld for Prompt tuning equal across all methods?
> > > >
> > > > A8. Thank you for raising this point. In our setup, prompt tuning freezes the entire backbone and only learns task-specific prompts. For the MetaWorld tasks, the pretrained agent has **limited prior knowledge of the unseen tasks**, and updating only the prompts provides insufficient capacity to meaningfully adapt the policy. As a result, the different finetuning objectives (SFT, RFT, QP, etc.) all operate on the same, severely constrained parameter subset, and converge to the same performance level. We will clarify this in the text.

---

> > > > > ### Author Response · Authors · 2025-11-21
> > > > >
> > > > > > Q5. Line 320 mentions that PPO and CQL lack inductive priors, what does that mean?
> > > > >
> > > > > A9. Thank you for pointing out this ambiguity.
> > > > > By saying that PPO and CQL “lack inductive priors”, we mean that, in our setup, these methods are not explicitly anchored to given demonstrations: they optimize value-based objectives directly on the few-shot finetuning data, which can lead to high-variance value estimates and less constrained policy updates in the low-data regime. In contrast, SFT impose a strong imitation prior by explicitly matching expert actions, which regularizes the policy toward demonstrated behavior and stabilizes finetuning. We will rephrase this sentence to state explicitly.
> > > > >
> > > > > > Q6. Line 323 mentions that switching from Adapters to Decorator for PPO yields 3% improvement, however the average score of Adapters for PPO is larger than the one of Decorator, am I missing something?
> > > > >
> > > > > A10. Thank you for catching this and giving us the chance to clarify.
> > > > >
> > > > > From L321: “Notably, in certain tasks—such as MetaWorld with Adaptor finetuning parameter—the choice of finetuning algorithm can yield greater performance improvements than merely increasing finetuning parameter size. For example, with PPO, switching the finetuning parameter from Adaptor to Decorator results in a modest 3% performance gain, whereas changing the algorithm to QP-DPO yields a 10\% improvement.” The statement at L323 refers **specifically to the MetaWorld setting**, not to the average performance across all environments. In MetaWorld, PPO with the **Decorator** configuration slightly outperforms PPO with the **Adaptor** configuration (from **454.91** to **468.47**, i.e., ~3% improvement). We will explicitly qualify this claim.
> > > > >
> > > > > >Q7. The final reached reward for the sparse setting (Figure 3) is around the same (sometimes even higher) than for the dense setting, why? What is the delay in the reward? For the same number of update steps I would expect massive differences if there was a substantial delay in the reward as shown in [1], which does not seem to be the case for e.g. PPO or CQL. [1] RUDDER: Return Decomposition for Delayed Rewards, Arjona-Medina et al., NeurIPS 2019
> > > > >
> > > > > > Q8. Any intuition as to why PPO trained on expert data is better in sparse reward settings than in dense reward settings?
> > > > >
> > > > > A11.  We appreciate this thoughtful question and are happy to clarify both the setup and the intuition.
> > > > >
> > > > > In our experiments, the *sparse* setting is constructed by taking each existing trajectory and setting **all intermediate step rewards to 0**, assigning the **entire cumulative return to the final timestep only**. This mirrors common RL finetuning practice in LLM reasoning, where each generated trajectory receives a **single terminal reward**. Importantly, the **total return per trajectory is preserved**; only its temporal distribution changes.
> > > > >
> > > > > This differs from the setting in RUDDER [1], which studies **online MDPs with delayed rewards** and focuses on how reward redistribution affects *credit assignment during exploration*. By contrast, our setting is **offline and few-shot**, with *fixed trajectories* and no exploration. Thus, we could not expect the same dramatic degradation as in online delayed-reward scenarios.
> > > > >
> > > > > In the offline setting, PPO-style updates rely on **advantage estimates** derived from Q- or value-function approximations. In the *dense* case, these estimates depend on many noisy intermediate rewards; with **very limited trajectories**, this can amplify approximation error for mid-trajectory state–action pairs and lead to misestimated advantages. In the *sparse* case, the reward signal is concentrated at the terminal step, so the value target is driven by a **single, well-defined terminal return**. This can, sometimes, **reduce noise in the advantage estimates** under data scarcity, especially when the trajectories are already near-expert. Thus, PPO trained on **expert data** may even perform slightly **better in the sparse setting**, because the advantage estimates are less corrupted by noisy or poorly estimated intermediate rewards.

---

> > > > > > ### Comment · Reviewer_EYuE · 2025-11-26
> > > > > >
> > > > > > Thank you for the additional experiments and the clarification of some confusions. I enumerate some open points below.
> > > > > >
> > > > > > **Inconsistent results and takeaways**
> > > > > >
> > > > > > Thank you for toning down the claims on the takeaways. They mostly address my concerns, up to the claim on "monotonic improvements", please correct this one as the trend is clearly not monotonic for some methods, not even typically.
> > > > > >
> > > > > > **Conceptual framing**
> > > > > >
> > > > > > Thank you for your explanation, as you correctly stated you frame your experimental design as metaRL setting without trying any metaRL algorithms. My suggestion here would be not to draw the connection to metaRL, or maybe add support that fine-tuning is usually on-par or better [1].
> > > > > >
> > > > > > [1] Mandi et al., On the Effectiveness of Fine-tuning Versus Meta-reinforcement Learning, NeurIPS 2022
> > > > > >
> > > > > > With respect to the TGA definition: "Generative Agents" has been used in prior work to define language-based agents, therefore this can lead to confusion. The term generative is generally unnecessary, one could argue each policy is generative regardless of being a Transformer or not as it generates an action. To reduce the potential for confusion, I would refrain from using the TGA terminology.
> > > > > >
> > > > > > **Positioning**
> > > > > >
> > > > > > My main point in bringing the related works, especially Schweighofer et al., up was to suggest that the metrics introduced therein could be used to quantify the quality and state-action coverage for your datasets. It would be interesting to see how much the results align with those in Schweighofer et al., and where your datasets position on those metrics.
> > > > > >
> > > > > > **Dense vs sparse rewards**
> > > > > >
> > > > > > > In the offline setting, PPO-style updates rely on advantage estimates derived from Q- or value-function approximations. In the dense case, these estimates depend on many noisy intermediate rewards; with very limited trajectories, this can amplify approximation error for mid-trajectory state–action pairs and lead to misestimated advantages.
> > > > > >
> > > > > > This is quite unintuitive, a dense reward should provide a stronger signal to learn from. Those environments usually come with dense rewards, why would they be noisy? Can the authors iteration on that?
> > > > > > After revisiting the results it is actually the case that most RL settings perform worse on the dense reward.
> > > > > >
> > > > > > If the authors address these remaining concerns/questions, I am willing to raise my score.

---

> > > > > > > ### Author Response · Authors · 2025-11-27
> > > > > > >
> > > > > > > Thank you for your thoughtful review and your kind support. We have carefully addressed all of your continual comments below and thoroughly revised the manuscript, with all changes highlighted in red. We hope our responses satisfactorily resolve your concerns.
> > > > > > >
> > > > > > > >Q9. Thank you for toning down the claims on the takeaways. They mostly address my concerns, up to the claim on "monotonic improvements", please correct this one as the trend is clearly not monotonic for some methods, not even typically.
> > > > > > >
> > > > > > > A12. Thank you for this clarification. We agree and will remove the “monotonic” phrasing from the takeaways. In the revised version, we will state more cautiously that:
> > > > > > >
> > > > > > > “Increasing the number of finetuning trajectories does **not** universally lead to performance improvements; the effect depends on the choice of finetuning algorithm and environment.”
> > > > > > >
> > > > > > > >Q10. Thank you for your explanation, as you correctly stated you frame your experimental design as metaRL setting without trying any metaRL algorithms. My suggestion here would be not to draw the connection to metaRL, or maybe add support that fine-tuning is usually on-par or better [1].
> > > > > > >
> > > > > > > A13. Thank you for pointing us to this relevant work.
> > > > > > >
> > > > > > > We will revise to more carefully connect our setting to prior results on **fine-tuning vs. meta-RL**, and we will explicitly cite [1], who show that fine-tuning can be competitive with or superior to meta-RL in many settings. This supports our choice to focus on **finetuning-based adaptation methods.**
> > > > > > >
> > > > > > > [1] Mandi et al., On the Effectiveness of Fine-tuning Versus Meta-reinforcement Learning, NeurIPS 2022
> > > > > > >
> > > > > > > >Q11. With respect to the TGA definition: "Generative Agents" has been used in prior work to define language-based agents, therefore this can lead to confusion. The term generative is generally unnecessary, one could argue each policy is generative regardless of being a Transformer or not as it generates an action. To reduce the potential for confusion, I would refrain from using the TGA terminology.
> > > > > > >
> > > > > > > A14. Thank you for this helpful suggestion.
> > > > > > >
> > > > > > > We will consistently use **“Transformer-based Agents (TAs)”** throughout the paper, and we will also clarify early on that these TAs **cast decision-making as sequence modeling over trajectories**, in line with Decision Transformer–style formulations.

---

> > > > > > > > ### Author Response · Authors · 2025-11-27
> > > > > > > >
> > > > > > > > >Q12. My main point in bringing the related works, especially Schweighofer et al., up was to suggest that the metrics introduced therein could be used to quantify the quality and state-action coverage for your datasets. It would be interesting to see how much the results align with those in Schweighofer et al., and where your datasets position on those metrics.
> > > > > > > >
> > > > > > > > A15. We sincerely thank the reviewer for this insightful suggestion. Following your recommendation, we computed the TQ and SACo metrics introduced by Schweighofer et al.[2] for the finetuning datasets used in our study. For each benchmark, we evaluate and average the metrics across the 5 meta-test tasks (50 trajectories per task) under both Expert and Medium demonstration regimes. For TQ and SACo normalization, we use the full collected dataset (Medium + Expert) as the reference buffer (e.g., 200 trajectories for HalfCheetahDir, 2000 for MetaWorld). Unlike [2], we do not sweep random seeds for dataset construction, and therefore we report single-seed values alongside performance results at traj_num = 50.
> > > > > > > >
> > > > > > > > | Env | Quality | Best algo (traj=50) | TQ | SACo |
> > > > > > > > | --- | --- | --- | --- | --- |
> > > > > > > > | AntDir | Expert | QP-SFT | 1.00 |  0.24 |
> > > > > > > > | AntDir | Medium | QP-DPO | 0.41 | 0.27 |
> > > > > > > > | HalfCheetahDir | Expert | QP-DPO | 0.99 | 0.27 |
> > > > > > > > | HalfCheetahDir | Medium | QP-SFT | 0.53 | 0.13 |
> > > > > > > > | HalfCheetahVel | Expert | GRPO | 0.99 |  0.50 |
> > > > > > > > | HalfCheetahVel | Medium | QP-SFT | 0.21 | 0.09 |
> > > > > > > > | MetaWorld | Expert | QP-DPO | 1.00 |  0.04 |
> > > > > > > > | MetaWorld | Medium | QP-SFT | 0.34 | 0.06 |
> > > > > > > >
> > > > > > > > These results reveal several trends consistent with [2]:
> > > > > > > >
> > > > > > > > - **Expert vs. Medium Regimes.** Expert datasets consistently occupy the **high-TQ region** (TQ ≈ 1.0) across environments, whereas Medium datasets exhibit substantially lower TQ (≈0.21–0.53). SACo values are generally low due to the limited number of finetuning trajectories (50), with particularly low coverage observed in MetaWorld.
> > > > > > > > - **Low TQ + Low SACo region is most challenging.**
> > > > > > > >
> > > > > > > >     HalfCheetahVel-Medium (TQ = 0.21, SACo = 0.09) and HalfCheetahDir-Medium (TQ = 0.53, SACo = 0.13) correspond to **low-quality, low-coverage** datasets and indeed yield the weak downstream performance, matching the “low TQ, low coverage” failure region identified in [2].
> > > > > > > >
> > > > > > > > - **Coverage benefits RFT, in line with [2].**
> > > > > > > >
> > > > > > > >     When SACo is relatively high (e.g., HalfCheetahVel-Expert), RFT-style methods (such as GRPO) gain more performance—mirroring Schweighofer et al.’s finding that unconstrained off-policy Deep Q-Network family requires datasets with high SACo to find a good policy.
> > > > > > > >
> > > > > > > >
> > > > > > > > Overall, mapping our datasets into the TQ–SACo landscape proposed by [2] demonstrates that:
> > > > > > > >
> > > > > > > > (1) **Trajectory quality is a key determinant of finetuning effectiveness**, and
> > > > > > > >
> > > > > > > > (2) **State–action coverage enhances the effectiveness of RFT methods**, whereas datasets that are simultaneously low in TQ and SACo are challenging for all algorithms.
> > > > > > > >
> > > > > > > > We will incorporate these results and discussion into the revision, including the above table and summary interpretation.
> > > > > > > >
> > > > > > > > [2] A dataset perspective on offline reinforcement learning, Schweighofer et al., CoLLAs 2022

---

> > > > > > > > > ### Author Response · Authors · 2025-11-27
> > > > > > > > >
> > > > > > > > > >Q13. This is quite unintuitive, a dense reward should provide a stronger signal to learn from. Those environments usually come with dense rewards, why would they be noisy? Can the authors iteration on that? After revisiting the results it is actually the case that most RL settings perform worse on the dense reward.
> > > > > > > > >
> > > > > > > > > A16. We thank the reviewer for pushing on this point. To clarify: the **noise is not in the environment’s dense reward function**, but in the **value/advantage estimation** that arises when doing TD-style bootstrapping from *very limited offline data, specially for the such low SACo metric.*
> > > > > > > > >
> > > > > > > > > We will revise the explanation along the following lines:
> > > > > > > > >
> > > > > > > > > 1. **Dense vs. sparse in the offline few-shot regime.**
> > > > > > > > >
> > > > > > > > >     In standard *online and offline* RL, dense rewards are clearly beneficial for credit assignment and exploration. In our setting, however, we operate in an **offline, few-shot regime** (e.g., 50 trajectories), with no further interaction. Here, RFT-style updates rely heavily on **function approximation + TD bootstrapping.** With very limited data, the learned Q- or value function can be inaccurate for many intermediate states. TD updates then **propagate these approximation errors backward** along the trajectory, compounding bias in the advantage estimates—especially in the dense case, where every step contributes a TD target.
> > > > > > > > >
> > > > > > > > > 2. **Why sparse can look better here.**
> > > > > > > > >
> > > > > > > > >     In our “sparse” construction, we preserve the **same total return**, but assign it only to the final timestep. This shifts the update toward a more **Monte Carlo–like regime**, where:
> > > > > > > > >
> > > > > > > > >     - returns for expert / near-expert trajectories are relatively stable and low-variance, and
> > > > > > > > >     - advantage estimates are driven more directly by the **actual episode return**, rather than by iterated bootstrapping through an imperfect value network at every intermediate step.
> > > > > > > > >
> > > > > > > > >     In this low-data offline setting, this can make the effective signal **less biased and more robust (relative).**
> > > > > > > > >
> > > > > > > > >
> > > > > > > > > Thus, when we previously referred to “noisy” dense rewards, what we meant is **estimation noise introduced by TD bootstrapping with a poorly estimated value function on limited data**, not that the environment’s dense reward itself is inherently noisy.
> > > > > > > > >
> > > > > > > > > We will update the manuscript to reflect this more accurate and nuanced explanation.

---

> > > > > > > > > > ### Comment · Reviewer_EYuE · 2025-11-27
> > > > > > > > > >
> > > > > > > > > > Thank you for the additional results and interpretation, I believe these additions add more rigor and depth to the paper, therefore I am increasing my score. It is a bit unfortunate that the takeaways are not as consistent as one would have hoped, but I am overall satisfied with the revisions and can recommend acceptance.
> > > > > > > > > >
> > > > > > > > > > Also, thank you for the more in-depth interpretation on dense vs sparse rewards, I am not entirely convinced by your argument, but the empirical results show that there is some latent variable that causes the gap, which could potentially be the caused by approximation errors.
> > > > > > > > > >
> > > > > > > > > > PS: You might want to remove the "generative" term from the title and the section 2.1 header for consistency.

---

> > > > > > > > > > > ### Author Response · Authors · 2025-11-28
> > > > > > > > > > >
> > > > > > > > > > > We sincerely thank you for your positive feedback and for raising your score. We are delighted to hear that our additional experiments and revisions have addressed your concerns and that you recommend acceptance. Your constructive comments throughout this process have significantly enhanced the rigor and depth of our manuscript.
> > > > > > > > > > >
> > > > > > > > > > > Regarding the dense vs. sparse rewards, we appreciate your insights. We agree that the interplay between reward density, data scarcity, and approximation error is nuanced. We believe our empirical findings provide a valuable foundation for understanding these dynamics, paving the way for future work to further disentangle these factors.
> > > > > > > > > > >
> > > > > > > > > > > We will also remove "Generative" from the title and Section 2.1 as suggested to ensure consistency. Thank you again for your valuable guidance; we look forward to contributing to the community's progress.

---

### Official Review · Reviewer_CvWD · 2025-10-31

**Soundness:** 3
**Presentation:** 3
**Contribution:** 3
**Rating:** 6
**Confidence:** 4

**Summary:**

This paper presents a comprehensive study on Reinforcement Fine-Tuning (RFT) for Transformer-based Generative Agents (TGAs) within meta-reinforcement learning (meta-RL) settings. The authors explore various finetuning algorithms, parameter configurations, and their interplay in adapting TGAs to new tasks. They propose a lightweight enhancement combining Supervised Fine-Tuning (SFT) and RFT, providing empirical evidence of its effectiveness across multiple meta-RL environments. The study’s results show that RFT can improve performance, especially in few-shot and sparse reward settings, and the proposed QP-based methods outperform traditional approaches like SFT.

**Strengths:**

1.The paper systematically evaluates different RFT methods, providing valuable insights into the trade-offs between various finetuning strategies.

2.The introduction of a lightweight enhancement that combines SFT and RFT is a notable innovation, and the proposed QP-based finetuning methods offer a promising direction for improving model performance in meta-RL settings.

3.The authors conduct extensive experiments across multiple environments (MuJoCo, MetaWorld) with varying dataset qualities and sizes, demonstrating the robustness of their methods.

4.The research addresses a practical challenge in adapting large pre-trained models to real-world tasks with limited data, an important topic given the increasing use of Transformer-based agents.

**Weaknesses:**

1.While the paper highlights the success of RFT in non-RL domains, the motivation for applying RFT to TGAs in meta-RL environments lacks a strong theoretical justification. The analogy to non-RL models feels speculative, without clearly explaining the structural or optimization similarities that would make RFT effective in RL settings. A more principled explanation would strengthen the rationale.

2.The introduction of QP (Q-guided Policy Optimization) is promising, but it requires more clarification. The explanation of how QP combines RL with SFT is not immediately clear, and the potential advantages and challenges of QP are not adequately addressed. A more concise and focused summary would help readers better understand its benefits.

3.The paper focuses on models with up to 40M parameters but does not address the applicability or efficiency of RFT in larger models, such as those with billions of parameters, which are common in current large language models. A discussion on scalability would provide a more complete picture of the method's practical limitations.

4.The conclusion summarizes the contributions well but could benefit from explicitly discussing the broader implications of the proposed RFT method for meta-RL and real-world applications. Additionally, acknowledging potential limitations, such as the method's dependency on smaller model scales, would offer a more balanced view.

**Questions:**

1.The paper mentions a “...lightweight improvement DP...” that integrates the advantages of SFT and RFT. However, the term "DP" is not clearly defined.

2.Can the authors provide a more detailed theoretical explanation for why RFT should benefit TGAs in meta-RL, beyond analogy with non-RL models?

3.How does the QP method combine RL and SFT? What are its key advantages and potential challenges in practical applications?

4.Given the increasing size of modern language models, how scalable is the proposed RFT method to models with billions of parameters, and what computational challenges might arise?

5.Could the authors discuss the broader impact of their RFT method on meta-RL and real-world applications, and highlight any potential limitations?

---

> ### Author Response · Authors · 2025-11-21
>
> Thank you for your thoughtful review and your kind support. We have carefully addressed all of your comments below and revised the manuscript, with all changes highlighted in red.
>
> > W1. While the paper highlights the success of RFT in non-RL domains, the motivation for applying RFT to TGAs in meta-RL environments lacks a strong theoretical justification. The analogy to non-RL models feels speculative, without clearly explaining the structural or optimization similarities that would make RFT effective in RL settings. A more principled explanation would strengthen the rationale.
>
> >Q2. Can the authors provide a more detailed theoretical explanation for why RFT should benefit TGAs in meta-RL, beyond analogy with non-RL models?
>
> A1. We thank the reviewer for this thoughtful question. Our motivation grounds in several concrete structural and optimization parallels:
>
> 1. **Structural alignment.** Both LLMs and TGAs are *autoregressive Transformers* conditioned on sequential context—tokens in NLP vs. $(s, a, r)$ tuples in RL. In both cases, the model learns to predict the next element in the sequence based on the history tokens, so policy optimization can be written as **sequence-level RL** over outputs. This shared structure makes RFT-style updates directly applicable to TGAs.
>     - **LLMs with RLHF/RFT:** model $p(y_t | y_{<t})$ over text tokens, optimizing for human-preference rewards.
>     - **TGAs:** model $p(a_t | s_{\le t}, a_{<t}, r_{\le t})$ over actions conditioned on trajectories and are optimized for **task rewards or return-to-go**, operating in **simulated control environments** and evaluated by task-specific reward metrics rather than preference alignment.
> 2. **Optimization alignment.** As detailed in Sec. 3.1, objectives such as DPO and GRPO can all be expressed as regularized policy-gradient surrogates with KL or ratio-based penalties, matching the mathematical form of RLHF/RFT objectives used in reasoning LLMs. The same gradient structure (log-probability ratios weighted by advantage or preference) provides a principled basis for expecting similar stability and reward-shaping benefits in TGAs.
> 3. **Meta-RL setting and empirical support.** In meta-RL, TGAs must rapidly adapt to new tasks from limited offline trajectories, akin to the way LLM adjust to human preferences from a small number of demonstrations. Here, pure SFT provides a strong imitation prior but no explicit policy improvement, while RFT-type objectives introduce value- or advantage-based corrections to move beyond the behavior policy. Our QP-based results (Figs. 1–5) empirically support this view, showing consistent gains across data quality, sparsity, and model scale.
>
> We will clarify this theoretical alignment more explicitly in the revised Section 2.1.
>
> > W2. The introduction of QP (Q-guided Policy Optimization) is promising, but it requires more clarification. The explanation of how QP combines RL with SFT is not immediately clear, and the potential advantages and challenges of QP are not adequately addressed. A more concise and focused summary would help readers better understand its benefits.
>
> > Q3. How does the QP method combine RL and SFT? What are its key advantages and potential challenges in practical applications?
>
> A2. We appreciate the request for clarification. QP is explicitly designed to **bridge the stability of SFT** and the **policy-improvement capability of RFT**.
>
> **How QP combines SFT and RL:**
>
> As defined in Eqs. (13)–(14), QP augments the supervised loss (SFT or DPO) with a Q-guided policy-improvement term. For example, for QP-SFT:
>
> $L_{\text{QP-SFT}}(\theta) = \mathbb{E}\_{(s,a)} \big[ \lVert a - \pi_\theta(a|s)\rVert^2 - \alpha Q_\phi(s,\pi_\theta(s)) \big].$
>
> The first term is the standard imitation loss (SFT), while the second term encourages the policy to choose actions with higher predicted return under $Q_\phi$. Thus, QP keeps the **SFT objective as an anchor** but adds a **value-based correction** that nudges the policy beyond the behavior policy, in the spirit of conservative policy improvement.
>
> **Key advantages (also reflected in experiments):**
>
> 1. **Stability:** The supervised term keeps updates close to the data distribution, mitigating the instability and divergence often seen in pure offline RL under distributional shift.
> 2. **Robust performance:** Across tasks and parameter configurations, QP variants (QP-SFT / QP-DPO) consistently match or exceed the strongest SFT and RL baselines, indicating that the hybrid objective provides a reliable improvement path rather than a brittle heuristic.
>
> **Practical challenges:**
>
> 1.  it requires tuning $\alpha$ to balance imitation and value guidance;
> 2.  it depends on reasonably accurate Q-estimates, which can be harder to obtain in extremely sparse- or ill-defined-reward settings. We discuss these issues and their empirical manifestations in Sec. 6.
>
> We will revise Sec. 3.1 to present this hybrid formulation and its trade-offs more succinctly and explicitly.

---

> > ### Author Response · Authors · 2025-11-21
> >
> > >W3. The paper focuses on models with up to 40M parameters but does not address the applicability or efficiency of RFT in larger models, such as those with billions of parameters, which are common in current large language models. A discussion on scalability would provide a more complete picture of the method's practical limitations.
> >
> > >Q4. Given the increasing size of modern language models, how scalable is the proposed RFT method to models with billions of parameters, and what computational challenges might arise?
> >
> > A3. We agree that scalability is an important concern and appreciate the opportunity to clarify this.
> >
> > In the current paper, we already study **scaling trends** across TGAs from **5M to 193M parameters** (Figure 5, Table 2), observing that the relative behavior of finetuning strategies is largely preserved as capacity grows, with gains limited primarily by data scarcity rather than algorithmic instability.
> >
> > The use of smaller models helps make the experimentation **tractable**—particularly given the large number of configurations explored across data regimes, reward sparsity, and algorithm variants. This design also ensures that our work remains **reproducible** and **accessible**, which is essential for the broader research community, especially for those without large-scale computational resources. We believe this approach provides **foundational** insights, which are highly relevant and informative for scaling RFT to larger models and more complex settings.
> >
> > To further evaluate the method’s performance with larger-scale models and more complex environments, we conducted additional experiments using the **LIBERO simulation benchmark** [1]. LIBERO is specifically designed to test **complex reasoning and long-horizon decision-making** in embodied tasks. It includes four challenging task suites:
> >
> > - **LIBERO-Spatial**: Focuses on spatial relations and object positioning
> > - **LIBERO-Object**: Involves reasoning over diverse object categories
> > - **LIBERO-Goal**: Requires understanding and generalizing goal semantics
> > - **LIBERO-Long (LIBERO-10)**: Comprises extended, sequential tasks that demand multi-step reasoning and planning
> >
> > For these experiments, we use **Dita [2]**, a TGA model with **334M parameters** (221M trainable), which is fine-tuned for **40,000 steps** using **SFT** and our proposed **QP-SFT** approach. Dita is fine-tuned on **8× NVIDIA RTX 4090 GPUs** over **3 days**. The results show that **QP-SFT consistently outperforms SFT**, particularly on **long-horizon tasks**, where reasoning and planning demands are most significant:
> >
> > |  | Spatial | Object | Goal | Long |
> > | --- | --- | --- | --- | --- |
> > | SFT | 84.2% | 96.3% | 85.4% | 63.8% |
> > | QP | 89.0% | 95.4% | 87.8% | 67.8% |
> >
> > These results demonstrate that **QP-SFT** continues to provide **strong improvements** on larger models and more complex environments, further validating the method’s scalability and practical utility.
> >
> > Regarding **billion-parameter scales**, our method is in principle **computationally comparable to standard RFT/RLHF pipelines**: QP adds a lightweight Q-network and value-guided term on top of a standard SFT-style objective, so training and memory cost grow approximately linearly with model size. However, we acknowledge two practical challenges that we will state explicitly in the revision:
> >
> > 1. **Data and adaptation:** few-shot meta-adaptation becomes more delicate as model capacity grows, increasing the risk of overfitting when finetuning data are limited.
> > 2. **Compute and memory:** compared to pure SFT, QP introduces extra forward/backward passes through the Q-network, which can be significant at billion-parameter scale and could be mitigated by parameter-efficient tuning and shared backbones.
> >
> > [1] LIBERO: Benchmarking Knowledge Transfer for Lifelong Robot Learning
> >
> > [2] Dita: Scaling Diffusion Transformer for Generalist Vision-Language-Action Policy

---

> > > ### Author Response · Authors · 2025-11-21
> > >
> > > > W4. The conclusion summarizes the contributions well but could benefit from explicitly discussing the broader implications of the proposed RFT method for meta-RL and real-world applications. Additionally, acknowledging potential limitations, such as the method's dependency on smaller model scales, would offer a more balanced view.
> > >
> > > > Q5. Could the authors discuss the broader impact of their RFT method on meta-RL and real-world applications, and highlight any potential limitations?
> > >
> > > A4. We appreciate this suggestion. In the revised version, we will clarify that RFT—and QP in particular—provides a **general framework for adapting Transformer-based decision models in settings with limited or no environment interaction**, which is common in offline meta-RL and many real-world applications (e.g., robotics, autonomous driving, and safety-critical domains where online exploration is expensive or risky). In such scenarios, collecting a small set of high-quality expert trajectories is often far more practical than running large-scale online RL. By combining supervised imitation with value-guided policy improvement, QP helps bridge **offline RL, imitation learning, and LLM-style post-training** for embodied and multi-modal agents.
> > >
> > > We will also explicitly acknowledge the **limitations** of our current study:
> > >
> > > - Our experiments are conducted on **medium-scale models** and **static offline datasets**; we have not yet validated QP at **billion-parameter scale** or in fully interactive RLHF-style pipelines.
> > > - The effectiveness of RFT/ QP depends on the **quality of value estimation**: QP requires access to reasonably informative reward signals and a learned Q-function, which may be challenging in settings with extremely sparse, noisy, or poorly specified rewards.
> > >
> > > These broader implications and limitations will be incorporated into the conclusion and discussion to provide a more balanced and forward-looking perspective.
> > >
> > > > Q1. The paper mentions a “...lightweight improvement DP...” that integrates the advantages of SFT and RFT. However, the term "DP" is not clearly defined.
> > >
> > > A5. Thank you for catching this ambiguity. The phrase “lightweight improvement DP” is a typo. It should read “lightweight improvement QP”, referring to our proposed Q-guided Policy (QP) method that integrates the advantages of SFT and RFT. We will correct this terminology throughout the paper to avoid confusion.

---

> ### Comment · Reviewer_CvWD · 2025-11-26
>
> Thank you for your detailed reply.
>
> The response addressed most of my concerns. I will keep my positive rating and respect the AC’s final decision.

---

> > ### Author Response · Authors · 2025-11-26
> > **Thanks to Reviewer CvWD**
> >
> > Dear Reviewer CvWD,
> >
> > We are grateful that your concerns have been addressed mostly, and we sincerely appreciate your constructive feedback throughout the review process. We will carefully revise the manuscript to incorporate the corresponding suggestions provided in your review.
> >
> > Thanks again for your time and thoughtful comments!
> >
> > Kind regards,
> >
> > Paper15946 Authors

---

### Official Review · Reviewer_RG77 · 2025-11-01

**Soundness:** 3
**Presentation:** 3
**Contribution:** 2
**Rating:** 6
**Confidence:** 3

**Summary:**

This paper investigates how to fine-tune a Decision Transformer-style generative agent on a new task with RL in a few-shot way, as existing works mainly investigate supervised fine-tuning of such a generative agent. They compare several existing methods and a newly proposed method called QP which adds an additional term of maximizing the Q-value of the policy to either SFT or DPO. They conduct extensive experimental analysis on Mujoco and MetaWorld to investigate how different fine-tuning algorithm choices and adaptation methods influence the fine-tuning performance, which provide useful insights for future work on RL fine-tuning of generative agents.

**Strengths:**

1. The paper is clearly motivated with a research gap, i.e., RL fine-tuning of generative agents, to fill.
2. The paper thoroughly investigate many possible RL ways for fine-tuning, and different ways parameter-efficient fine-tuning methods.
3. The experiments extensively investigate many problem setting and algorithm choices that may influence the performance of RL fine-tuning.

**Weaknesses:**

1. My main concern is with the significance of novelty of this paper. It's more like a benchmarking paper instead of proposing a new idea. The QP method proposed by the authors is more like a direct extension of the QT (Hu et al. 2024) algorithm to a multi-task setting, which is limited in originality.
2. As discussed by the authors, this paper only considers moderate-size models on relatively simple benchmarks like Mujoco locomotion and MetaWorld. Whether the lessons learned from these benchmarking results can be extended to larger-scale models on more realistic tasks or not remains unknown.

**Questions:**

1. Which checkpoint's performance is reported for each method? The last one after training for a fixed amount of steps or the one with the best evaluation performance?
2. Why not do "real" PPO learning? Is it because you want to do few-shot adaptation?
3. The CQL regularization term in equation 11 seems to be inverse?

---

> ### Author Response · Authors · 2025-11-21
>
> Thank you for your thoughtful review and your kind support. We have carefully addressed all of your comments below and thoroughly revised the manuscript, with all changes highlighted in red. We hope our responses satisfactorily resolve your concerns.
>
> > W1. My main concern is with the significance of novelty of this paper. It's more like a benchmarking paper instead of proposing a new idea. The QP method proposed by the authors is more like a direct extension of the QT (Hu et al. 2024) algorithm to a multi-task setting, which is limited in originality.
>
> A1. We thank the reviewer for this important concern. Our intention is not to present a pure benchmark, but to deliver a **conceptual and methodological study** of RFT for Transformer-based decision agents, with QP as a minimal but principled extension.
>
> **Novel empirical question and cross-domain investigation:** RFT has been developed primarily for LRMs, yet its **transferability to Transformer-based generative agents (TGAs)**—which act in *sequential decision-making* environments rather than text domains—has not been systematically examined. Our work is, to our knowledge, the **first systematic study** of whether and when RFT-style updates that succeed in LRMs also benefit TGAs in meta-RL settings, which is a necessary step toward general-purpose agents.
>
> **Methodological extension beyond QT:** While QP is conceptually related to QT, it is **not** a mere multi-task restatement. QP integrates Q-value guidance directly into *policy-level finetuning objectives* (Eqs. 13–14), explicitly bridging **supervised imitation (SFT/DPO)** and **reinforcement-based policy improvement** in the few-shot offline regime. This design is motivated by the observed complementary strengths of SFT and RFT in our experiments. Moreover, we provide **new theoretical support**—including a bound on the deviation from SFT (Theorem E.4 and Corollary E.5).
>
> **Breadth and interpretive depth:** Beyond proposing QP, we conduct a **comprehensive empirical analysis** across (i) four finetuning parameter configurations, (ii) seven finetuning algorithms, (iii) two meta-RL benchmarks, and (iv) ablations on data quality, data quantity, reward sparsity, and model scale. This reveals previously unreported patterns in offline meta-RL settings, such as RFT methods outperforming SFT under expert data but degrading on medium-quality data.
>
> **Clarifying observed behavior:** While the focus of this paper is empirical, we **do not merely report numbers**. Throughout Sections 5 and 6, we provide **reasoned analyses and ablation studies** to explain key observations, such as the performance tradeoffs between SFT and RFT under different data regimes, and the role of Q-function guidance in stabilizing policy updates.
>
> In summary, our contribution lies in (i) extending RFT analysis to a new class of Transformer-based decision agents, (ii) introducing a theoretically grounded Q-guided hybrid finetuning objective, and (iii) providing the first systematic empirical characterization of RFT in this setting for practical deployment.

---

> > ### Author Response · Authors · 2025-11-21
> >
> > > W2. As discussed by the authors, this paper only considers moderate-size models on relatively simple benchmarks like Mujoco locomotion and MetaWorld. Whether the lessons learned from these benchmarking results can be extended to larger-scale models on more realistic tasks or not remains unknown.
> >
> > A2. We appreciate the reviewer’s concern and agree that scalability is a critical aspect to address.
> >
> > While our base agents are relatively small, we **evaluate across a range of Transformer-based generative agents (TGAs)**, from 5M to 193M parameters (as shown in Figure 5 and Table 2). This allows us to capture **scaling trends** and assess whether the performance characteristics of different finetuning strategies remain consistent at larger scales.
> >
> > The use of smaller models helps make the experimentation **tractable**—particularly given the large number of configurations explored across data regimes, reward sparsity, and algorithm variants. This design also ensures that our work remains **reproducible** and **accessible**, which is essential for the broader research community, especially for those without large-scale computational resources. We believe this approach provides **foundational** insights, which are highly relevant and informative for scaling RFT to larger models and more complex settings.
> >
> > To further evaluate the method’s performance with larger-scale models and more complex environments, we conduct additional experiments using the **LIBERO simulation benchmark** [1]. LIBERO is specifically designed to test **complex reasoning and long-horizon decision-making** in embodied tasks. It includes four challenging task suites:
> >
> > - **LIBERO-Spatial**: Focuses on spatial relations and object positioning
> > - **LIBERO-Object**: Involves reasoning over diverse object categories
> > - **LIBERO-Goal**: Requires understanding and generalizing goal semantics
> > - **LIBERO-Long (LIBERO-10)**: Comprises extended, sequential tasks that demand multi-step reasoning and planning
> >
> > For these experiments, we use **Dita [2]**, a TGA model with **334M parameters** (221M trainable), which is fine-tuned for **40,000 steps** using **SFT** and our proposed **QP-SFT** approach. Dita is fine-tuned on **8× NVIDIA RTX 4090 GPUs** over **3 days**. The results show that **QP-SFT consistently outperforms SFT**, particularly on **long-horizon tasks**, where reasoning and planning demands are most significant:
> >
> > |  | Spatial | Object | Goal | Long |
> > | --- | --- | --- | --- | --- |
> > | SFT | 84.2% | 96.3% | 85.4% | 63.8% |
> > | QP-SFT | 89.0% | 95.4% | 87.8% | 67.8% |
> >
> > These results demonstrate that **QP-SFT** continues to provide **strong improvements** on larger models and more complex environments, further validating the method’s scalability and practical utility. We will include these additional experiments in Appendix F and discuss their implications for future work involving larger-scale TGAs.
> >
> > [1] LIBERO: Benchmarking Knowledge Transfer for Lifelong Robot Learning
> >
> > [2] Dita: Scaling Diffusion Transformer for Generalist Vision-Language-Action Policy
> >
> > > Q1. Which checkpoint's performance is reported for each method? The last one after training for a fixed amount of steps or the one with the best evaluation performance?
> >
> > A3. We report the best-performing checkpoint for each method, selected based on the evaluation return on the held-out meta-test tasks, rather than the final iteration. This approach ensures a fair comparison across methods with potentially different convergence dynamics and is consistent with standard practices in meta-RL evaluation, where performance is typically measured at the point of optimal generalization.
> >
> > >Q2. Why not do "real" PPO learning? Is it because you want to do few-shot adaptation?
> >
> > A4. Our study focuses on offline few-shot adaptation, where agents must improve from a fixed set of trajectories without additional environment interaction. Real PPO requires on-policy sampling, which is infeasible in this setup. While the PPO algorithm is highly effective in online settings, its applicability is limited in strictly offline contexts. However, we still leverage the PPO loss for gradient computation in our framework to benefit from its stability and optimization properties.
> >
> > >Q3. The CQL regularization term in equation 11 seems to be inverse?
> >
> > A5. Thank you for pointing this out. We will correct the formulation. In Equation 11, the regularization term with parameter $\alpha$ penalizes the Q-value of the policy when it exceeds the Q-value of the behavior policy, thereby preventing overestimation of OOD actions.

---

### Author Response · Authors · 2025-11-21

# Summary

**We would like to thank all the reviewers for their thoughtful suggestions on our submission, and appreciate that the reviewers have multiple positive impressions of our work, including:**

- **A clear motivation and well-identified research gap** in exploring reinforcement finetuning (RFT) for generative agents (RG77, RqYc).
- **A systematic and comprehensive evaluation**, covering diverse RFT and SFT methods, multiple parameter-efficient finetuning configurations, and extensive ablations across environments, data regimes, reward structures, and model scales (RG77, CvWD, EYuE).
- **Insightful comparative analysis** that sheds light on the trade-offs among RFT strategies and their practical implications for few-shot meta-RL adaptation (CvWD).
- **A promising hybrid approach (QP)** that unifies supervised and reinforcement finetuning and demonstrates robust performance across varied settings (CvWD, EYuE).
- **Strong experimental breadth** across MuJoCo and MetaWorld tasks with varying dataset qualities and sizes, highlighting robustness of the methods (CvWD).
- **Relevance to practical real-world constraints**, especially the challenge of adapting large pretrained Transformer-based agents with limited data (CvWD).
- **Conceptual ambition and topical significance**, connecting reinforcement-based alignment with multi-task/offline RL adaptation for large Transformer-based agents (RqYc).

We summarize below the key revisions we will incorporate into the manuscript. **All corresponding clarifications, analyses, and additional results will be integrated into the revised submission.**

**Introduction, Related Works, and Conclusions:**

- We will clarify the **motivation, novelty, and research gap** addressed by the paper, explicitly positioning our work within the landscape of RFT for decision-making agents (RG77, CvWD, RqYc).
- We will expand the discussion of the **broader impact of RFT**, its relevance to real-world embodied or robotic applications, and its **potential limitations** (CvWD).
- All typographical issues noted in the introduction will be corrected (CvWD).
- We will integrate additional **related work**, including recent studies on data quality, PEFT methods, and multi-task RL, to better contextualize our contributions (EYuE).
- We will provide a clearer justification for evaluating RFT in the **offline meta-RL** setting and explain its relevance to domains where online interaction is infeasible (RqYc).

**Methods:**

- We will refine the mathematical presentation of baseline methods and correct all notational inaccuracies (RG77, EYuE).
- We will more clearly articulate the **design, intuition, and advantages** of the proposed QP method, including its connection to regularized policy improvement (CvWD).
- We will provide a **formal and precise definition of Transformer-based generative agents**, distinguishing them from LLM-based agents and clarifying their operational domain (EYuE, RqYc).

**Experiments:**

- We will incorporate **additional large-scale experiments on the LIBERO benchmark**, demonstrating QP’s applicability to more complex environments and larger models (RG77, CvWD).
- We will provide full experimental details, including: the backbone architecture and model scales, pretraining datasets and number of trajectories, finetuning dataset sizes, evaluation protocol and checkpoint selection criteria (RG77, EYuE, RqYc).
- We will **revise and tighten the key takeaways** to avoid overstatement, ensuring they reflect the empirical results with appropriate nuance (EYuE).
- We will include additional **Q-learning ablations**, including multi-action sampling for QP updates (EYuE).
- We will add deeper analysis of **counterintuitive behaviors**, such as: the identical performance of prompt tuning on MetaWorld, PPO performance in sparse vs. dense reward settings. These analyses will provide clearer insights into algorithmic behavior (EYuE, RqYc).

**Once again, we sincerely thank all reviewers for their detailed feedback and constructive suggestions. We will integrate all corresponding discussions, analyses, and additional experimental results into the revised manuscript.**

---

### Comment · Area_Chair_5Nsq · 2025-11-25

Dear Reviewers

Thank you for your time and help for reviews.
The author-reviewer discussion due is in one week. If you have not done yet, please review the authors' rebuttal for the paper under your evaluation and engage in discussion with authors.

Thank you again.
Best,

Area Chair

---

### Meta-Review · Area_Chair_fVrj · 2026-01-13

**Summary:**

Reviewers generally agreed that the paper offers a timely and systematic empirical study of reinforcement fine-tuning (RFT) for Transformer-based decision agents, an area less explored than supervised fine-tuning, with broad experimental coverage and a lightweight hybrid extension (QP) that combines SFT and RFT (RG77; CvWD; EYuE). The main concerns shaping the decision were initially unclear positioning (benchmark vs. method vs. analysis), conceptual and terminological ambiguities (e.g., “generative agents,” meta-RL framing), and questions about empirical consistency, scalability, and the justification of offline RFT (EYuE; RqYc). After rebuttal, most reviewers converged on viewing the work as a useful empirical and methodological contribution, with remaining issues largely about scope and framing rather than technical soundness.

**Reviewer Concerns:**

Most major concerns were addressed during the rebuttal. Overstated or inconsistent takeaways were corrected by softening claims, removing incorrect “monotonic” assertions, and adding clearer analyses of dense vs. sparse rewards (EYuE). Conceptual clarity improved through explicit positioning of the paper as a systematic empirical study with a minimal Q-guided extension, along with clearer motivation for offline RFT and tightened terminology (CvWD; EYuE). Scalability concerns were partially alleviated by added LIBERO experiments at 334M parameters, though very large-scale regimes remain future work (RG77; CvWD). Reproducibility issues around datasets, model scale, checkpoints, and baselines were clarified, resolving most factual objections (RG77; EYuE). One reviewer remained unconvinced by the offline RFT premise and continued to be an outlier (RqYc).

**Reviewer Scores:**

(RG77) would remain around 6, as novelty concerns persist but scalability and methodological clarity improved.

(CvWD) would remain around 6, having indicated that their main concerns were addressed and maintaining a positive stance.

(EYuE) explicitly increased their score to an accept-level (~6–7) after additional analyses, clarifications, and toned-down claims.

(RqYc) would likely remain low (~2), as their fundamental objections to the offline RFT framing and contribution scope were not fully alleviated despite detailed rebuttal.

---

### Decision · Program_Chairs · 2026-01-26

Accept (Poster)